# VALUE-ALIGNED WORLD MODEL REGULARIZATION FOR MODEL-BASED REINFORCEMENT LEARNING

## ABSTRACT

Model-based reinforcement learning (MBRL) aims to construct world models for imagined interactions to enable efficient sampling. Based on training strategy, current mainstream algorithms can be categorized into two types: maximum likelihood and value-aware world models. The former adopts structured Recurrent/Transformer State-Space Models (RSSM/TSSM) to capture environmental dynamics but may overlook task-relevant features. The latter focuses on decision-critical states by minimizing one-step value evaluations, but it often obtains sub-optimal performance and is difficult to scale. Recent work has attempted to integrate these approaches by leveraging the strong priors of pre-trained large models, though at the cost of increased computational complexity. In this work, we focus on combining these two approaches with minimal modifications. We empirically demonstrate that the key to their integration lies in: RSSM/TSSM ensuring the lower bound of the world model, while value awareness enhances the upper bound[1]. To this end, we introduce a value-alignment regularization term into the maximum likelihood world model learning, promoting task-aware feature reconstruction while modeling the stochastic dynamics. To stabilize training, we propose a warm-up phase and an adaptive weight mechanism for value-representation balance. Extensive experiments across 46 environments from the Atari 100k and DeepMind Control Suite benchmarks, covering both continuous and discrete action control tasks with visual and proprioceptive vector inputs, show that our algorithm consistently boosts existing MBRL methods performance and convergence speed with minimal additional code and computational complexity.

## 1 INTRODUCTION

In recent years, deep reinforcement learning (DRL) has achieved remarkable progress across various domains, including game playing(Vinyals et al., 2019), robotic control(Ju et al., 2022) and large model fine-tuning(Guo et al., 2025), driven by trial-and-error mechanism. However, the extensive samples required for training has limited DRL's deployment in real-world applications. To address this, model-based reinforcement learning (MBRL) has emerged as a promising solution, gaining considerable attention within the research community. The core idea of MBRL is the introduction of a world model, which, by modeling environment dynamics, reduces the need for frequent real interactions and facilitates efficient sampling. Based on different world model training strategies, current MBRL algorithms can be broadly classified into two types: maximum likelihood(Hafner et al., 2019a; Burchi & Timofte, 2025) and value-aware world models(Farahmand et al., 2017; Voelcker et al., 2022). The former adopts variational inference to directly model environment dynamics with RSSM/TSSM, while the latter incorporates value functions to emphasize task-relevant feature reconstruction. However, value-aware models have empirically struggled with suboptimal performance and scaling challenges, leading mainstream algorithms to primarily adopt the maximum likelihood approach, while the development of value-aware world models has progressed more slowly.

Dreamer(Hafner et al., 2019a; 2020; 2025), a pioneering work in maximum likelihood algorithms, successfully applies MBRL across various domains. The training process consists of two key stages: 1) world model training and 2) behavior model training, as shown in Fig.1(a). During the world model phase, the agent interacts with the real environment and trains the world model using collected

---

[1]The detailed definition of the upper and lower bounds can be found in Appendix J.

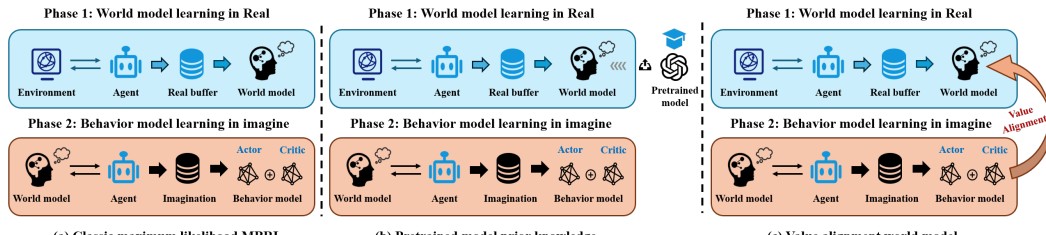

Figure 1: Classic maximum likelihood MBRL algorithm workflow and variants.

real trajectories. In the behavior model phase, the agent interacts solely with the learned world model and trains the behavior model using imagined trajectories. By alternating between these two stages, Dreamer achieves strong performance with minimal real interaction and substantial virtual imagination, significantly enhancing sampling efficiency. However, in this framework, the world model and behavior model training are often independent, leading to a misalignment between their objectives and causing task-relevant features to be overlooked. To address this, recent works, as shown in Fig.1(b), have introduced the prior knowledge of pre-trained large models to guide the world model's focus on significant information. For example, (Zhang et al., 2025a) uses object detection to prioritize decision-relevant target areas, while DreamVLA(Zhang et al., 2025b) enhances spatial reconstruction through depth-based 3D knowledge and semantic segmentation. Although incorporating pre-trained large models improves performance, the associated high computational complexity limits training efficiency. Additionally, the misalignment between pre-trained models and real environments poses risks to model performance.

From the above perspectives, we conclude two key limitations in current maximum likelihood world model methods: **1) Redundancy of input states:** In particular, visual inputs often contain substantial redundancy and the maximum likelihood loss treats each pixel equally, which may hinder critical information prediction. **2) Misalignment with task knowledge:** Due to the misalignment between training objectives, a good world model does not necessarily translate into a good policy. Therefore, it is crucial to identify features with task-specific knowledge. For example, in DMC Control tasks, accurately predicting pose is essential, while in Atari games, reconstructing the scene is more critical. To this end, we propose Value-aligned World Model(as shown in Fig.1(c)), which bridges world model and behavior model training through a value-alignment regularization term. On one hand, the value network reflects the environment's reward distribution, enabling the model to identify interest regions with high value fluctuations and achieve task alignment. On the other hand, value alignment term does not require additional prior knowledge or increase computational complexity, making it more convenient and efficient for deployment compared to pre-trained large models.

**Our Contribution.** In this paper, we address the challenges of input redundancy and task misalignment in current maximum likelihood world model algorithms. To this end, we propose Value-aligned World Model, a novel and effective MBRL algorithm that bridges the gap between world model and behavior model learning through value alignment. Specifically, we introduce a **Value-alignment regularization term (Var)** into the maximum likelihood world model optimization, allowing the world model to not only focus on modeling environmental dynamics but also prioritize the reconstruction of states with high value sensitivity. To ensure training stability, we design a warm-up phase and a value-representation adaptive weight mechanism, which prevent instability during the early stages of value learning and balance the maximum likelihood loss with the value-alignment regularization term, respectively. In practice, we apply our approach to two classic methods, DreamerV3(Hafner et al., 2025) and STORM(Zhang et al., 2023), and conduct extensive experiments across 26 environments from the Atari 100k benchmark(Bellemare et al., 2013) and 20 environments from the DMC Control benchmark(Tassa et al., 2018), covering both continuous and discrete action control tasks with visual and proprioceptive vector inputs. The experimental results show that our algorithm significantly improves the performance of existing MBRL baselines with faster convergence. Specifically, on the Atari 100k benchmark, our algorithm improves DreamerV3's average performance from 1.10 to 1.34 and its median performance from 0.58 to 1.00. This demonstrates that the proposed value-alignment regularization term consistently enhances model performance across various environments, rather than yielding large improvements in only a few extreme cases. Furthermore, our algorithm is best viewed as a plug-and-play module, requiring only a few lines of code to integrate into existing maximum likelihood methods, with minimal additional computational complexity.

## 2 RELATED WORK

Generally, most mainstream MBRL algorithms follow a two-stage training process: world model learning and behavior model learning. Depending on the strategy used to train the world model, MBRL algorithms can be further categorized into two types:

**Maximum likelihood world models** (Seo et al., 2023; Micheli et al., 2024) aim to accurately predict environmental dynamics from historical observations and actions, minimizing prediction errors via maximum likelihood estimation. PlaNet(Hafner et al., 2019b) introduces the Recurrent State-Space Model (RSSM), using recurrent neural networks (RNNs) and Variational Autoencoders (VAE)(Kingma & Welling, 2013) to model the world in latent space. Dreamer(Hafner et al., 2019a) builds on RSSM by incorporating an actor-critic framework that imagines behavior within the world model. DreamerV2(Hafner et al., 2020) optimizes this approach by replacing Gaussian latents with discrete categorical latents, improving stochastic dynamics representation. DreamerV3(Hafner et al., 2025) introduces structural and training modifications, enabling stable learning across various domains without hyperparameter tuning. Recent works have explored replacing RNN-based world models with Transformer architectures, incorporating self-attention mechanisms. TWM(Robine et al., 2023) proposes the Transformer State-Space Model (TSSM), treating states, actions, and rewards as independent tokens for dynamic modeling, while STORM(Zhang et al., 2023) integrates states and actions into a single token, enhancing training efficiency. More recently, DIAMOND(Alonso et al., 2024) introduced diffusion models for precise visual detail prediction, and TWISTER(Burchi & Timofte, 2025) applied Contrastive Predictive Coding in TSSM to model temporal dependencies. Despite these advancements, maximum likelihood world models still struggle with misalignment between the world model's training objectives and the policy optimization goal. Additionally, the need for each state precise prediction in maximum likelihood estimation limits the model's ability to effectively reconstruct task-relevant states, hindering its applicability in complex environments.

**Value-aware world models**, as the name suggests, aim to guide the world model with the value function to minimize the one-step value estimation error. The concept of Value-Aware Model Learning (VAML) was first introduced by (Farahmand et al., 2017), and IterVAML(Farahmand, 2018) was subsequently developed to iteratively optimize the policy and mitigate the "max-min" issue inherent in VAML. VaGraM(Voelcker et al., 2022) further enhances VAML by introducing Value-gradient weighted Model Learning, focusing the model on states that significantly influence the policy. More recently, CVAML(Voelcker et al.) introduces a variance correction term to address "overconfidence" in stochastic environments. While value-aware world models provide an intuitive approach to address the misalignment issue inherent in maximum likelihood world models, the instability of value estimation for out-of-distribution samples and the non-convexity of the VAML loss function make these algorithms susceptible to local optima during training. This, in turn, complicates their practical deployment and results in suboptimal performance compared to maximum likelihood-based methods. Moreover, these algorithms have not demonstrated strong empirical performance in complex, high-dimensional visual environments, such as Atari games.

Recent works have attempted to integrate these two approaches. For example, TEMPO(Yuan et al., 2023) introduces a bi-level framework, adding a meta-weighting network atop the maximum-likelihood model to generate sample weights that minimize task-aware model loss. While TEMPO shows promising results, the bi-level structure significantly increases computational complexity, inference time, and resource consumption, making practical deployment challenging. Other approaches, inspired by the rise of pre-trained large models, leverage their prior knowledge and generalization capabilities to replace value functions for decision-sensitive reconstruction. PSP(Hutson et al., 2024) incorporates a pre-trained segmentation model, enabling the world model to capture key environmental features. (Zhang et al., 2025a) assigns higher optimization weights to decision-relevant regions using object detection. DreamVLA(Zhang et al., 2025b) improves world model predictions by integrating 3D knowledge and semantic segmentation. While pre-trained large models improve performance, their high computational demands limit training efficiency. Additionally, the potential misalignment between pre-trained models and tasks complicates effective world model optimization.

In this work, we aim to seamlessly integrate maximum likelihood and value-aware world model learning with minimal modifications, building on existing algorithms. We introduce a value-alignment regularization term into the maximum likelihood world model, directing the model's focus to value-sensitive regions. To balance the environmental dynamics prediction loss with value-alignment

regularization, we propose a warm-up phase and an adaptive weight mechanism to mitigate value instability and avoid local optima, common in VAML. With minimal code changes, our algorithm can be easily integrated into existing maximum likelihood MBRL methods.

# 3 PRELIMINARIES

**Reinforcement Learning:** We consider a Markov Decision Process (MDP)(Puterman, 1990) defined as a tuple $(S, A, r(s, a), P(s'|s, a), \gamma)$, where $S$ and $A$ represent the state and action spaces, $r(s, a)$ is the reward function, $P(s'|s, a)$ denotes the state transition dynamics and $\gamma \in (0, 1)$ is the discount factor. The objective of reinforcement learning is to optimize the cumulative reward over time

**Model-based Reinforcement Learning:** MBRL introduces a world model $\alpha$ in the latent space to represent the environment dynamics $P(z'|z, a)$, where $z$ denotes the latent state representation $s$ under a given encoder. We consider the MBRL paradigm of learning through imagination, which involves three iteratively repeated phases: experience collection, world model learning, and policy learning. Specifically, the agent learns the policy behavior entirely within the world model, with real interaction trajectories used exclusively for world model training.

# 4 METHODS

In this section, we first explore the motivation for combining maximum likelihood world model and value-aware world model, providing empirical insights into how these approaches mutually enhance each other. Using DreamerV3(Hafner et al., 2025) as an example, we then demonstrate the integration of the value-alignment regularization term into maximum likelihood world model optimization.

## 4.1 MOTIVATION FOR COMBINING MAXIMUM LIKELIHOOD AND VALUE-AWARE LEARNING

To integrate maximum likelihood and value-aware methods effectively, the first step is to analyze the strengths and limitations of each approach. Starting with maximum likelihood world models, which typically use RSSM/TSSM as the core architecture in latent space, these models leverage structured variational inference to capture complex latent stochastic dynamics and generalize to unseen distributions. However, in optimization, these models minimize the prediction error between predictions and ground truth using maximum likelihood loss, without incorporating additional priors or constraints. This becomes problematic when model capacity is limited or inputs are highly redundant: it hampers the model's ability to capture task-relevant features and misaligns the objectives of world model training and policy optimization, ultimately reducing model performance.

Value-aware world models typically use standard RNNs to directly predict environmental dynamics. Compared to RSSM/TSSM, these models lack the ability to capture stochastic events and complex dynamics, often leading to local optima. In optimization, value functions guide the model to minimize one-step value estimation errors while reducing prediction errors in environmental dynamics. This constraint directs the model to focus on task-relevant features, addressing the misalignment issue. However, due to their simplified architecture and the non-convex nature of the VAML loss(Voelcker et al., 2022), these methods are difficult to implement and struggle to scale in complex environments.

Given the strengths and limitations, combining maximum likelihood and value-aware is a natural progression. The maximum likelihood method, with RSSM/TSSM at its core, ensures a stable lower bound, while value-aware learning enhances the upper bound. By integrating both architectural strengths and optimization strategies, we can achieve substantial performance improvements.

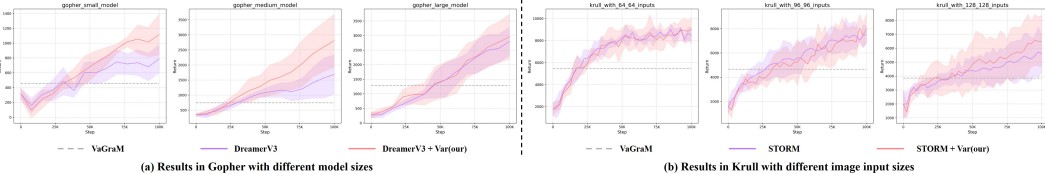

Figure 2: Experimental results across different model sizes and input dimensions.

As shown in Fig.2, we conducted a series of experiments to empirically validate the above analysis. Specifically, we compare the maximum likelihood algorithms, DreamerV3(Hafner et al., 2025) and STORM(Zhang et al., 2023), with the value-aware algorithm, VaGraM(Voelcker et al., 2022), as baselines within the Gopher and Krull visual games, which respectively evaluate the short-term and long-term planning capabilities. Fig.2(a) presents results across different world model capacities (input size: 64x64), with three settings: small (1M), medium (12M) and large (25M). The results indicate that with a larger world model capacity, the model better captures environmental dynamics. In this case, the world model can accurately reconstruct the next frame, with relatively minor performance improvement from the value-alignment regularization. However, with a smaller world model capacity, the model struggles to capture the dynamics, leading to a significant drop in performance. In this scenario, introducing value-alignment regularization helps the world model focus on reconstructing critical states, resulting in substantial performance improvements. Fig.2(b) presents the results with different image input sizes (world model capacity: 12M), using three configurations: 64x64, 96x96 and 128x128. The results demonstrate that as the input image size increases, more redundant information is introduced, and the maximum likelihood loss struggles to capture critical, task-relevant features, leading to performance degradation. However, the introduction of value-alignment regularization significantly alleviates this issue.

These two experiments demonstrate that for typical maximum likelihood world model algorithms (DreamerV3 and STORM), when model capacity is limited and input information is redundant, introducing value-alignment regularization to enhance task-awareness yields significant benefits. Across all experiments conducted, we observe that the performance of the value-aware algorithm, VaGraM, generally falls short compared to the maximum likelihood algorithms. This further underscores that, in MBRL, a powerful world model architecture (RSSM/TSSM) guarantees the lower bound of algorithm performance, while value-alignment awareness improves the upper bound, particularly in challenging deployment scenarios.

## 4.2 VALUE-ALIGNED WORLD MODEL LEARNING

The world model aims to capture environmental dynamics and state representations, enabling the imagination of future trajectories based on potential actions. Following DreamerV3(Hafner et al., 2025), we implement the world model using a Recurrent State-Space Model, parameterized as $\alpha$. Specifically, given an image observation $o_t$, we map it to a latent stochastic representation $z_t$ via an encoder network, which is a VAE with categorical latents. A temporal sequence model then predicts the next recurrent state $h_t$ based on the previous recurrent state $h_{t-1}$, latent representation $z_{t-1}$, and action $a_{t-1}$. Finally, the model state $s_t = \{h_t, z_t\}$, formed by concatenating $h_t$ and $z_t$, is used to predict the environment reward $r_t$, the episode continuation flag $c_t$, and to reconstruct the input $o_t$ via a decoder network. Specifically, the encoder and decoder use convolutional neural networks (CNNs) for image inputs and multilayer perceptrons (MLPs) for vector inputs. The sequence model is based on a recurrent neural networks (RNNs), while the dynamics, reward, and continuation predictors are implemented as MLPs. The components of the RSSM-based world model are illustrated below:

$$
\text{RSSM}
\begin{cases}
\text{Sequence model: } h_t = f_\alpha(h_{t-1}, z_{t-1}, a_{t-1}) \\
\text{Encoder Network: } z_t \sim q_\alpha(z_t|h_t, o_t) \\
\text{Dynamics Predictor: } \tilde{z}_t \sim p_\alpha(\tilde{z}_t|h_t) \\
\text{Reward Predictor: } \tilde{r}_t \sim p_\alpha(\tilde{r}_t|s_t) \\
\text{Continue Predictor: } \tilde{c}_t \sim p_\alpha(\tilde{c}_t|s_t) \\
\text{Decoder Network: } \tilde{o}_t \sim p_\alpha(\tilde{o}_t|s_t)
\end{cases}
\tag{1}
$$

**World Model Loss Function:** Given a batch size $B$ and sequence length $T$, with input observations $o_{1:T}$, actions $a_{1:T}$, rewards $r_{1:T}$, and episode continuation flags $c_{1:T}$, the world model is optimized end-to-end by minimizing the following loss function:

$$
L_{world} = \frac{1}{B \times T} \sum_{b=1}^{B} \sum_{t=1}^{T} \Big[ L_{pred} + L_{dyn} + \mathbb{1}_{ts>10^4} \cdot \beta_{var} L_{var} \Big]
\tag{2}
$$

The prediction loss $L_{\text{pred}}$ is computed using symlog squared loss to train the decoder network and reward predictor, while logistic regression is applied to train the continuation predictor. The dynamics loss $L_{\text{dyn}}$ is used to train the sequence model by minimizing the KL divergence between the predicted distribution $p_\alpha(\tilde{z}_t|h_t)$ and the next encoder representation $q_\alpha(z_t|h_t, o_t)$. In practice, $L_{\text{dyn}}$ utilizes the stop gradient operator $sg(\cdot)$ to prevent backpropagation of gradients. Additionally, a representation loss is introduced to encourage the encoder to learn more predictable state representations. To further enhance focus on suboptimal parts of the dynamics, clipping is applied. These two loss components: the prediction loss and dynamics loss, form the standard world model loss function in DreamerV3, as follows, with $\beta_{\text{dyn}} = 0.5$ and $\beta_{\text{rep}} = 0.1$:

$$
\begin{aligned}
L_{pred} &= -\ln p_\alpha(o_t|s_t) - \ln p_\alpha(r_t|s_t) - \ln p_\alpha(c_t|s_t) \\
L_{dyn} &= \beta_{dyn} \max\big(1, \text{KL}\big[\text{sg}(q_\phi(z_t|h_t, o_t)) \;||\; p_\phi(\tilde{z}_t|h_t)\big]\big) \\
&+ \beta_{reg} \max\big(1, \text{KL}\big[\; q_\phi(z_t|h_t, o_t) \;||\; \text{sg}(p_\phi(\tilde{z}_t|h_t))\big]\big)
\end{aligned}
\tag{3}
$$

The value-alignment loss $L_{\text{var}}$ acts as a regularization term to guide the optimization of the standard DreamerV3 world model, encouraging the model to focus on task-relevant, value-sensitive information during reconstruction. Unlike traditional value-aware world models, which explicitly influence the world model's dynamic representation learning through one-step value estimation errors or value-gradient weighting, we draw inspiration from perceptual loss in computer vision. We implicitly inject value-awareness into the world model via value alignment. To enhance generalization and mitigate the risk of local overfitting, we refrain from using the final sampled value scalar. Instead, we leverage the intermediate distribution output by the value network in DreamerV3(Hafner et al., 2025), applying KL divergence to enforce value alignment. Structurally, we follow the same design as the dynamics loss $L_{\text{dyn}}$, introducing the stop gradient operator $sg(\cdot)$ to stabilize the training process. The specific formulation is as follows:

$$
\begin{aligned}
L_{var} &= \beta_{dyn}\text{KL}\big[\text{sg}(V_\theta(v_t|s_t)) \;||\; V_\theta(\tilde{v}_t|\tilde{s}_t)\big] \\
&+ \beta_{reg}\text{KL}\big[\; V_\theta(v_t|s_t) \;||\; \text{sg}(V_\theta(\tilde{v}_t|\tilde{s}_t))\big]
\end{aligned}
\tag{4}
$$

Building on value-alignment loss $L_{var}$, we further introduce an indicator function $\mathbb{1}_{ts>10^4}$ and an adaptive weight $\beta_{\text{var}}$ to implement the warm-up phase and balance the trade-off between value alignment and dynamic representation loss. Specifically, we designate the first 10,000 training steps as the warm-up phase, during which value-alignment regularization is disabled to avoid training instability caused by inaccurate early-stage value network evaluations. Equally important is balancing the extent of value alignment with the dynamic representation loss, where the magnitude of the dynamic representation loss indicates the similarity between model's predictions and the real environment. We consider that value alignment is effective only when the world model's predictions closely match the real environment; if the gap is too large, value alignment may hinder learning. Therefore, our design prioritizes dynamic representation loss, followed by value-alignment regularization. In practice, we employ the inverse of the dynamic representation loss $L_{\text{dyn}}$ as the adaptive weight $\beta_{\text{var}}$. When the dynamic representation loss is large, indicating a significant discrepancy between the model's predictions and the ground truth, the weight of the value-alignment regularization is reduced, focusing learning on improving dynamic representations. Conversely, when the dynamic representation loss is small, suggesting that the model's predictions are closely aligned with the real environment, the weight of the value-alignment regularization increases, shifting focus toward achieving value-awareness.

$$
\begin{aligned}
\mathbb{1}_{ts>10^4} &= 1 \;\text{ if training steps} > 10^4, \;\text{ else } \; 0 \\
\beta_{var} &= 1 \,/\, max\big(1, \text{sg}\big(\text{KL}\big[q_\phi(z_t|h_t, o_t) \;||\; p_\phi(\tilde{z}_t|h_t)\big]\big)\big)
\end{aligned}
\tag{5}
$$

### 4.3 Agent behavior learning

Following DreamerV3(Hafner et al., 2025), both the critic and actor networks are trained using imagined trajectories generated by the world model. For environment interaction, actions are selected by sampling from the actor network without lookahead planning. In practice, both networks are implemented as MLPs, parameterized by $\theta$ and $\phi$, respectively.

$$\text{Critic Network:} \quad v_t \sim V_\theta(v_t|s_t) \qquad \text{Actor Network:} \quad a_t \sim \pi_\phi(a_t|s_t) \tag{6}$$

**Critic Learning:** In line with DreamerV3(Hafner et al., 2025), we estimate returns that incorporate rewards beyond the prediction horizon by computing bootstrapped $\lambda$-returns, which combine both predicted rewards and value estimates. The critic network is trained to predict the distribution of these $\lambda$-return estimates $R_t^\lambda$ by minimizing the maximum likelihood loss.

$$L_{critic} = \frac{1}{B \times T} \sum_{b=1}^{B} \sum_{t=1}^{T} -\ln p_\theta(R_T^\lambda|s_t) \quad R_t^\lambda = r_t + \gamma c_t\big((1-\lambda)v_t + \lambda R_{t+1}^\lambda\big) \tag{7}$$

**Actor Learning:** The actor network maximizes cumulative rewards using the REINFORCE(Williams, 1992) algorithm, with an added policy entropy loss to ensure sufficient exploration.

$$L_{actor} = \frac{1}{B \times T} \sum_{b=1}^{B} \sum_{t=1}^{T} -\text{sg}(A_t^\lambda)\log\pi_\phi(a_t|s_t) - \eta\text{H}\big[\pi_\phi(a_t|s_t)\big] \tag{8}$$

Here, $A_t^\lambda$ represents the advantage computed using normalized returns. To ensure stable learning, the returns are scaled using the exponentially moving average of the 5th and 95th percentiles of the batch.

$$A_t^\lambda = (R_t^\lambda - V_\theta(s_t))/\max(1,S) \quad S = \text{EMA}\big(\text{Per}(R_t^\lambda, 95) - \text{Per}(R_t^\lambda, 5), 0.99\big) \tag{9}$$

## 5 EXPERIMENTS

### 5.1 BENCHMARKS AND BASELINES

To rigorously assess our method, we evaluate on the following two well-established benchmarks:

**(1) The Atari 100k benchmark**(Kaiser et al., 2019) consists of 26 Atari games with discrete action controls, utilizing a budget of 400k environment frames, equivalent to approximately two hours of actual gameplay. Following (Burchi & Timofte, 2025), we choose RNN-based DreamerV3(Hafner et al., 2025), transformer-based TWM(Robine et al., 2023), IRIS(Micheli et al., 2022) and STORM(Zhang et al., 2023), as well as SimPLe(Kaiser et al., 2019), as baselines.

**(2) The DeepMind Control Suite**(Tassa et al., 2018) is divided into two components based on input types. The Proprio Control part consists of 18 continuous action tasks with proprioceptive vector inputs, using a budget of 500K environment steps. These tasks span classical control domains, ranging from locomotion to robotic manipulation, and feature both dense and sparse reward scenes. Following (Hafner et al., 2025), we select PPO(Schulman et al., 2017), DMPO(Abdolmaleki et al., 2018), D4PG(Barth-Maron et al., 2018) and DreamerV3(Hafner et al., 2025) as baselines. The Visual Control part comprises 20 continuous control tasks and a budget of 1M environment steps. Following (Hafner et al., 2025), we choose PPO(Schulman et al., 2017), SAC(Haarnoja et al., 2018), CURL(Laskin et al., 2020), DrQ-v2(Yarats et al., 2021) and DreamerV3(Hafner et al., 2025) as baselines.

### 5.2 RESULTS ON ATARI 100K

Tab.1 presents the quantitative results of applying value-alignment regularization (Var) to DreamerV3(Hafner et al., 2025) and STORM(Zhang et al., 2023) on the Atari 100k benchmark, while Fig.5 shows the training curves. To ensure fair comparison, we retrained both DreamerV3 and STORM using identical hyperparameters. Following previous work, we used human-normalized metrics to evaluate performance across 26 games, comparing mean and median scores. The results demonstrate consistent performance improvements: for DreamerV3, 24 out of 26 games showed improvements, with the average score increasing from 1.10 to 1.34 and the median from 0.58 to 1.00. Similarly, STORM improved in 24 games, with the average score rising from 1.14 to 1.36 and the median from 0.51 to 0.81. Notably, games such as KungFuMaster, Gopher, Qbert and Kangaroo, where small target characters are crucial, exhibited particularly significant performance gains.

Table 1: Quantitative results on the Atari 100k benchmark. We show average scores over 5 seeds.

| Game | Random | Human | SimPLe | TWM | IRIS | DreamerV3 | DreamerV3+Var (our) | STORM | STORM+Var (our) |
|------|--------|-------|--------|-----|------|-----------|---------------------|-------|-----------------|
| Alien | 227.8 | 7127.7 | 616.9 | 674.6 | 420.0 | 875.88 | 1233.2(↑40.8%) | 1054.3 | 1361.4(↑29.1%) |
| Amidar | 5.8 | 1719.5 | 74.3 | 121.8 | 143.0 | 143.7 | 185.4(↑29.0%) | 177.29 | 248.36(↑40.1%) |
| Assault | 222.4 | 742.0 | 527.2 | 682.6 | 1524.4 | 843.7 | 981.38(↑16.3%) | 715.9 | 752.55(↑5.1%) |
| Asterix | 210.0 | 8503.3 | 1128.3 | 1116.6 | 853.6 | 1102.5 | 1162.6(↑5.5%) | 1276.0 | 1535.0(↑20.3%) |
| BankHeist | 14.2 | 753.1 | 34.2 | 466.7 | 53.1 | 1072.0 | 1121.2(↑4.6%) | 1060.5 | 935.0(↓11.8%) |
| BattleZone | 2360.0 | 37187.7 | 4031.2 | 5068.0 | 13074.0 | 11138.0 | 12750.0(↑14.5%) | 7080.0 | 10140.0(↑43.2%) |
| Boxing | 0.1 | 12.1 | 7.8 | 77.5 | 70.1 | 80.3 | 87.4(↑8.9%) | 78.6 | 83.0(↑5.6%) |
| Breakout | 1.7 | 30.5 | 16.4 | 20.0 | 83.7 | 25.3 | 45.6(↑79.9%) | 20.88 | 26.43(↑26.6%) |
| ChopperCommand | 811.0 | 7387.8 | 979.4 | 1697.4 | 1565.0 | 1438.0 | 1826.0(↑27.0%) | 1768.0 | 1695.0(↓4.1%) |
| CrazyClimber | 10780.5 | 35829.4 | 62583.6 | 71820.4 | 59324.2 | 89900.0 | 81720.0(↓9.1%) | 47473.0 | 57335.0(↑20.8%) |
| DemonAttack | 152.1 | 1971.0 | 208.1 | 350.2 | 2034.4 | 223.9 | 227.2(↑1.5%) | 194.6 | 204.6(↑5.1%) |
| Freeway | 0.0 | 29.6 | 16.7 | 24.3 | 31.1 | 30.2 | 31.6(↑4.6%) | 29.7 | 32.0(↑7.8%) |
| Frostbite | 65.2 | 4334.7 | 236.9 | 1475.6 | 259.1 | 1628.0 | 347.9(↓78.6%) | 258.8 | 260.2(↑0.5%) |
| Gopher | 257.6 | 2412.5 | 596.8 | 1674.8 | 2236.1 | 1683.9 | 2807.0(↑66.7%) | 8551.0 | 13509.6(↑58.0%) |
| Hero | 1027.0 | 30826.4 | 2656.6 | 7254.0 | 7037.4 | 4994.4 | 9360.6(↑87.4%) | 12249.2 | 12574.0(↑2.7%) |
| Jamesbond | 29.0 | 302.8 | 100.5 | 362.4 | 462.7 | 332.0 | 542.0(↑63.3%) | 446.4 | 462.5(↑3.6%) |
| Kangaroo | 52.0 | 3035.0 | 51.2 | 1240.0 | 838.2 | 1529.2 | 3650.4(↑138.7%) | 1542.0 | 3322.6(↑115.4%) |
| Krull | 1598.0 | 2665.5 | 2204.8 | 6349.2 | 6616.4 | 8364.8 | 9821.4(↑17.4%) | 8360.1 | 8896.5(↑6.4%) |
| KungFuMaster | 258.5 | 22736.3 | 14862.5 | 24554.6 | 21759.8 | 16375.0 | 21075.0(↑28.7%) | 15760 | 26615.0(↑68.9%) |
| MsPacman | 307.3 | 6951.6 | 1480.0 | 1588.4 | 999.1 | 1947.0 | 1749.5(↓10.1%) | 1906.9 | 2417.3(↑26.8%) |
| Pong | -20.7 | 14.6 | 12.8 | 18.8 | 14.6 | 19.1 | 19.8(↑4.0%) | 20.6 | 20.2(↓1.9%) |
| PrivateEye | 24.9 | 69571.3 | 35.0 | 86.6 | 100.0 | 2331.2 | -115.6(↓104.9%) | 414.4 | 2584.7(↑523%) |
| Qbert | 163.9 | 13455 | 1288.8 | 3330.8 | 745.7 | 1223.5 | 2267.8(↑85.4%) | 2912.5 | 4243.4(↑45.7%) |
| RoadRunner | 11.5 | 7845.0 | 5640.6 | 9109.0 | 9614.6 | 9868.6 | 14704.0(↑49.0%) | 11523.0 | 13999.0(↑21.5%) |
| Seaquest | 68.4 | 42054.7 | 683.3 | 774.4 | 661.3 | 513.2 | 546.3(↑6.5%) | 441.4 | 430.0(↓2.6%) |
| UpNDown | 533.4 | 11693.2 | 3350.3 | 15981.7 | 3546.2 | 12679.2 | 18485.4(↑45.8%) | 6406.4 | 8982.6(↑40.2%) |
| Superhuman (↑) | 0 | N/A | 1 | 8 | 10 | 10 | 13(↑3) | 9 | 12(↑3) |
| Mean (↑) | 0.00 | 1.00 | 0.33 | 0.6 | 1.05 | 1.10 | 1.34(↑0.24) | 1.14 | 1.36(↑0.22) |
| Median (↑) | 0.00 | 1.00 | 0.13 | 0.51 | 0.29 | 0.58 | 1.00(↑0.42) | 0.51 | 0.81(↑0.30) |

## 5.3 RESULTS ON DEEPMIND CONTROL SUITE

Tab.2 presents the quantitative results of applying value-alignment regularization (Var) to DreamerV3(Hafner et al., 2025) on the DMC Suite benchmark. To ensure a fair comparison, DreamerV3 was retrained with identical hyperparameters for both input modalities. The results show consistent performance improvements across continuous control tasks: with visual inputs, 15 out of 20 tasks saw improvements, with the average score increasing from 792 to 827 and the median from 877 to 894; with vector inputs, performance improved in 13 out of 18 tasks, with the average score rising from 805 to 817 and the median from 881 to 901. Fig.4 shows the training curves for the DMC Suite benchmark. These results demonstrate that our approach accelerates the convergence of MBRL algorithms, especially in tasks like Pendulum Swingup and Walker Walk.

Table 2: Quantitative results on the DMC suite benchmark. We show average scores over 5 seeds.

| Task | PPO | SAC | CURL | DrQ-v2 | DreamerV3 | DreamerV3+Var | PPO | DDPG | DMPO | D4PG | DreamerV3 | DreamerV3+Var |
|------|-----|-----|------|--------|-----------|---------------|-----|------|------|------|-----------|---------------|
| Input types | Visual Image Inputs | | | | | | Proprioceptive Inputs | | | | | |
| Environment steps | 1M | 1M | 1M | 1M | 1M | 1M | 500K | 500K | 500K | 500K | 500K | 500K |
| Acrobot Swingup | 3 | 4 | 4 | 166 | 314 | 367(↑16.7%) | 6 | 100 | 103 | 124 | 261 | 295(↑13.1%) |
| Ball In Cup Catch | 829 | 176 | 970 | 928 | 953 | 967(↑1.5%) | 632 | 917 | 968 | 968 | 968 | 965(↓0.4%) |
| Cartpole Balance | 516 | 937 | 980 | 992 | 998 | 999(↑0.08%) | 523 | 997 | 999 | 999 | 997 | 999(↑0.2%) |
| Cartpole Balance Sparse | 881 | 956 | 999 | 987 | 1000 | 1000(0.00%) | 930 | 992 | 999 | 974 | 989 | 990(↑0.1%) |
| Cartpole Swingup | 290 | 706 | 771 | 863 | 866 | 865(↓0.2%) | 240 | 864 | 860 | 875 | 872 | 865(↓0.7%) |
| Cartpole Swingup Sparse | 1 | 149 | 373 | 773 | 520 | 756(↑45.6%) | 7 | 703 | 438 | 752 | 802 | 800(↓0.3%) |
| Cheetah Run | 95 | 20 | 502 | 716 | 917 | 916(↓0.1%) | 82 | 596 | 650 | 624 | 748 | 834(↑11.4%) |
| Finger Spin | 118 | 291 | 880 | 862 | 520 | 602(↑15.7%) | 18 | 775 | 769 | 823 | 536 | 537(↑0.2%) |
| Finger Turn Easy | 253 | 200 | 340 | 525 | 888 | 914(↑3.0%) | 281 | 499 | 620 | 612 | 889 | 892(↑0.3%) |
| Finger Turn Hard | 79 | 94 | 231 | 247 | 895 | 885(↓1.0%) | 106 | 313 | 495 | 421 | 975 | 977(↑0.2%) |
| Hopper Hop | 0 | 0 | 164 | 221 | 325 | 336(↑3.3%) | 0 | 36 | 68 | 80 | 236 | 238(↑1.0%) |
| Hopper Stand | 4 | 5 | 777 | 903 | 938 | 933(↓0.5%) | 3 | 484 | 549 | 762 | 862 | 910(↑5.5%) |
| Pendulum Swingup | 1 | 592 | 413 | 843 | 807 | 812(↑0.6%) | 1 | 767 | 834 | 759 | 805 | 852(↑5.8%) |
| Quadruped Run | 88 | 54 | 149 | 450 | 782 | 824(↑5.4%) | - | - | - | - | - | - |
| Quadruped Walk | 112 | 49 | 121 | 726 | 810 | 902(↑11.4%) | - | - | - | - | - | - |
| Reacher Easy | 487 | 67 | 689 | 944 | 924 | 961(↑4.0%) | 494 | 934 | 961 | 960 | 962 | 969(↑0.7%) |
| Reacher Hard | 94 | 7 | 472 | 670 | 759 | 797(↑4.9%) | 288 | 949 | 968 | 937 | 965 | 960(↓0.5%) |
| Walker Run | 30 | 27 | 360 | 539 | 688 | 764(↑11.0%) | 31 | 561 | 493 | 616 | 726 | 716(↓1.4%) |
| Walker Stand | 161 | 143 | 486 | 978 | 983 | 985(↑0.2%) | 159 | 965 | 975 | 947 | 967 | 976(↑0.9%) |
| Walker Walk | 87 | 40 | 822 | 768 | 960 | 961(↑0.2%) | 64 | 952 | 942 | 969 | 930 | 933(↑0.3%) |
| Task mean | 206 | 226 | 525 | 705 | 792 | 827(↑4.4%) | 215 | 689 | 705 | 733 | 805 | 817(↑1.5%) |
| Task median | 94 | 81 | 479 | 770 | 877 | 894(↑1.9%) | 94 | 771 | 801 | 792 | 881 | 901(↑2.3%) |

## 5.4 VISUALIZATION AND ANALYSIS OF IMAGINED TRAJECTORIES

Fig.3 illustrates the visualization of imagined trajectories generated by the world model. The top two rows show the imagined trajectories of STORM without value-alignment regularization, accompanied by a heatmap of decoder network sensitivity. The next two rows display the imagined trajectories with value-alignment regularization and a sensitivity heatmap after value-gradient weighting. The

bottom row presents the ground truth. The visual results highlight the benefits of value alignment in two key aspects: **(1) Single-frame prediction:** The original STORM(Zhang et al., 2023) algorithm often suffers from target disappearance, blurring and hallucinations. After value alignment, the weighted heatmap focuses more on foreground objects, avoiding irrelevant background features. **(2) Long-term sequence prediction:** The original STORM algorithm experiences significant divergence in later predictions due to accumulated errors. Value-alignment regularization, however, maintains better temporal consistency across the sequence. (For long-sequence visualizations, see Fig.7.)

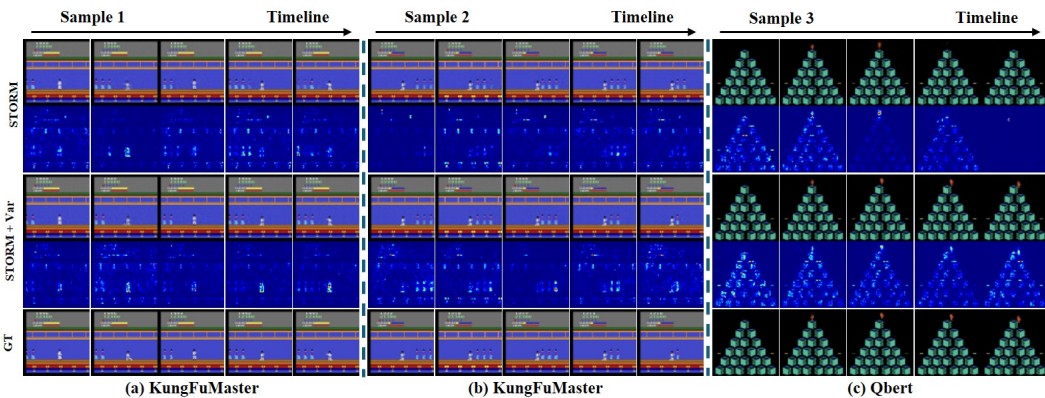

Figure 3: Imagined trajectories from the world model in KungFuMaster and Qbert games

## 5.5 ABLATION STUDY

We conduct an ablation study on the Atari 100k benchmark, using STORM as the baseline to evaluate the proposed adaptive weight mechanism. For comparison, we use a static weight of 0.5 as a control. Tab.3 presents the quantitative results across five environments: Alien, CrazyClimber, DemonAttack, BankHeist, and BattleZone. The results show that introducing adaptive weights to balance dynamic representation loss and value-alignment regularization improves world model optimization, stabilizes training, and enhances model performance. The training curves are provided in Fig.6.

Table 3: Ablation study on the adaptive weighting.

| Atari Games | Alien | CrazyClimber | DemonAttack | BankHeist | BattleZone |
|---|---|---|---|---|---|
| STORM | 1054.3 | 47473.0 | 194.6 | 1060.5 | 7080.0 |
| STORM + static weight | 987.9(↓6.3%) | 55096.0(↑16.1%) | 180.3(↓7.3%) | 656(↓38.1%) | 8480(↑19.8%) |
| STORM + adaptive weight | 1361.4(↑29.1%) | 57335.0(↑20.8%) | 204.6(↑5.1%) | 935.0(↓11.8%) | 10140(↑43.2%) |

We conduct tests on an NVIDIA 3090 GPU to evaluate the impact of value-alignment regularization on GPU memory and runtime. Tab.4 summarizes the effects on computational resources and training time. The results show that the computational overhead and training time introduced by value-alignment regularization are minimal, making their impact negligible relative to the overall algorithmic cost.

Table 4: Ablation study on additional computational resources and runtime.

| Methods | DreamerV3 | DreamerV3 + Var(our) | STORM | STORM + Var(our) |
|---|---|---|---|---|
| GPU Memory | 4560MB | 4626MB | 5806MB | 5864MB |
| total running time | 15.89h | 16.06h | 5.91h | 6.04h |
| Mean Score on Atari 100k | 1.10 | 1.34 | 1.14 | 1.36 |

## 6 CONCLUSION

In this work, we integrate maximum likelihood and value-aware approaches in model-based reinforcement learning, enhancing task-relevant feature reconstruction by incorporating value-awareness into maximum likelihood world model optimization. Specifically, we introduce a novel value-aligned world model that ensures a stable lower bound through the RSSM/TSSM architecture, while value-alignment regularization improves the upper bound. To stabilize training, we implement an adaptive weighting mechanism to balance dynamic representation loss with value-alignment regularization. Extensive experiments across 46 environments from the Atari 100k and DeepMind Control Suite benchmarks show that our approach consistently improves the performance and convergence speed of existing MBRL methods, with minimal additional complexity, particularly in complex, high-dimensional environments.

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

# A TRAINING CURVES ACROSS VARIOUS BENCHMARKS

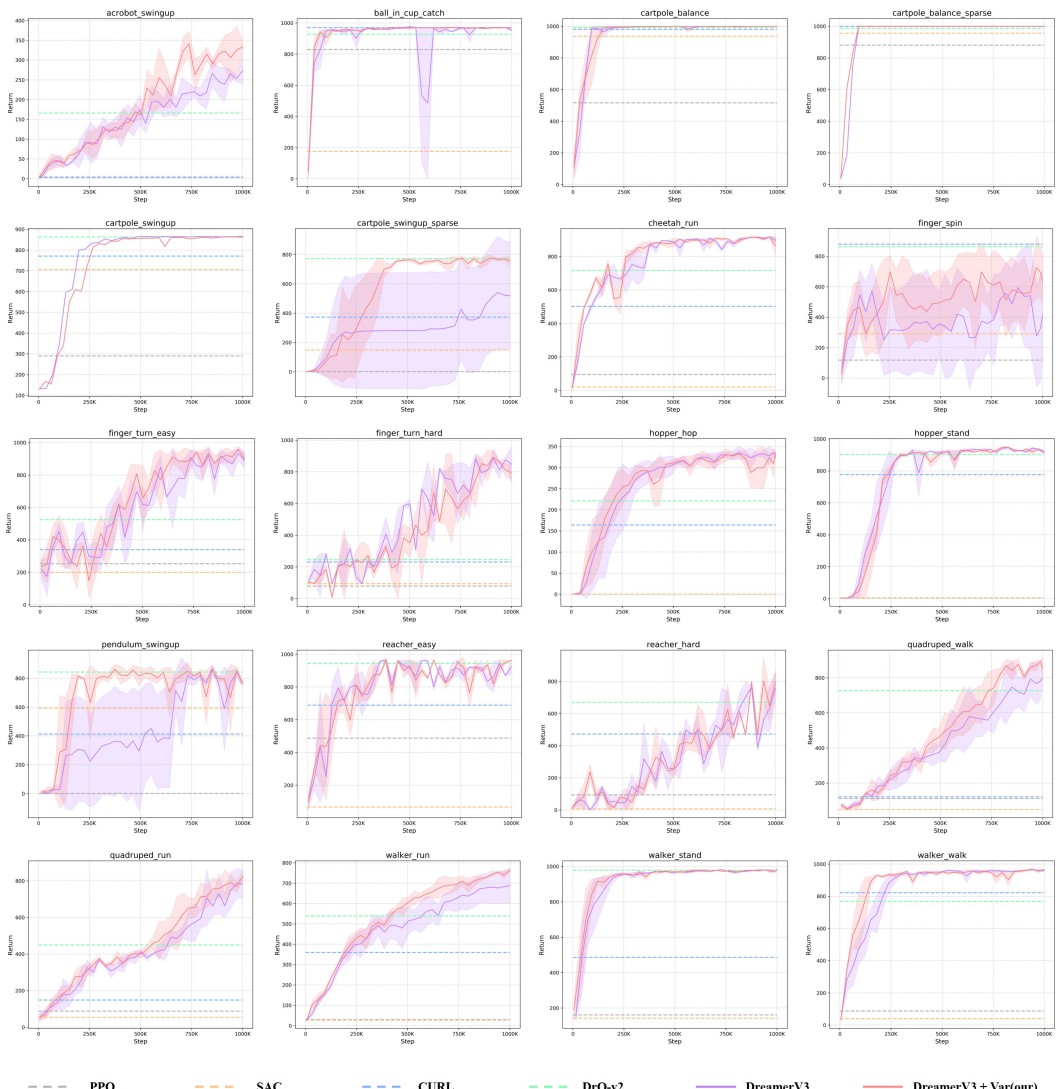

Figure 4: Training curve on DMC suite.

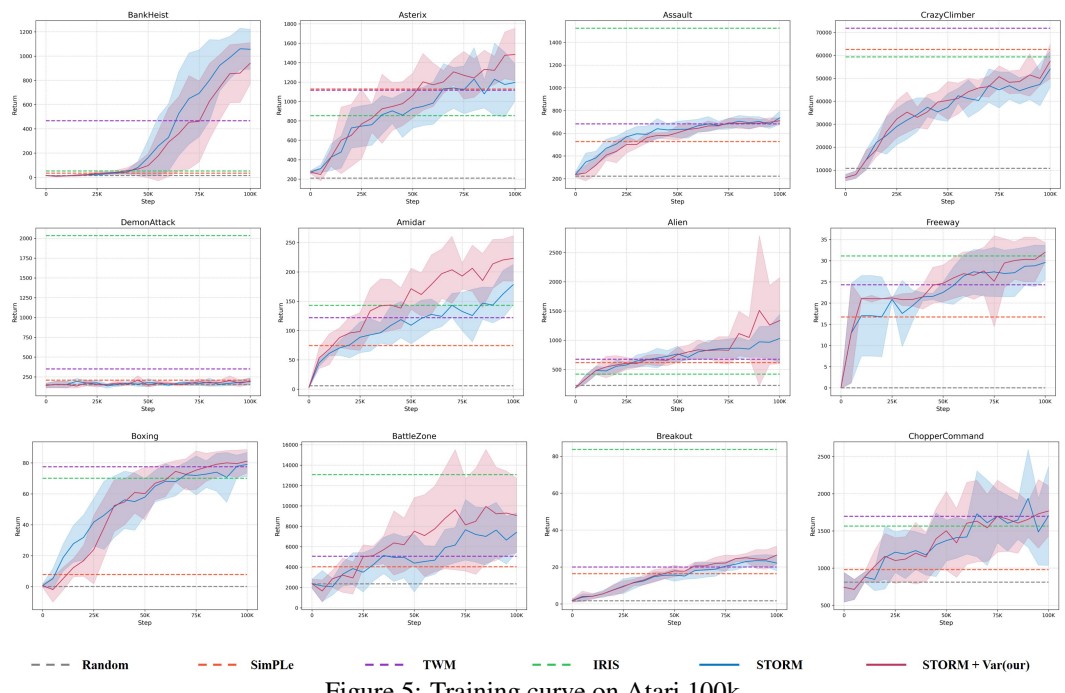

Figure 5: Training curve on Atari 100k.

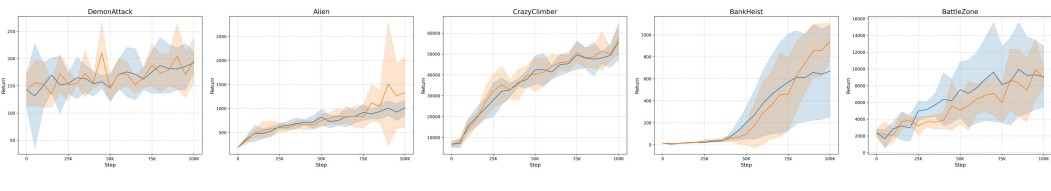

Figure 6: Training curve on static and adaptive weighting.

## B LONG-TERM IMAGINED TRAJECTORIES

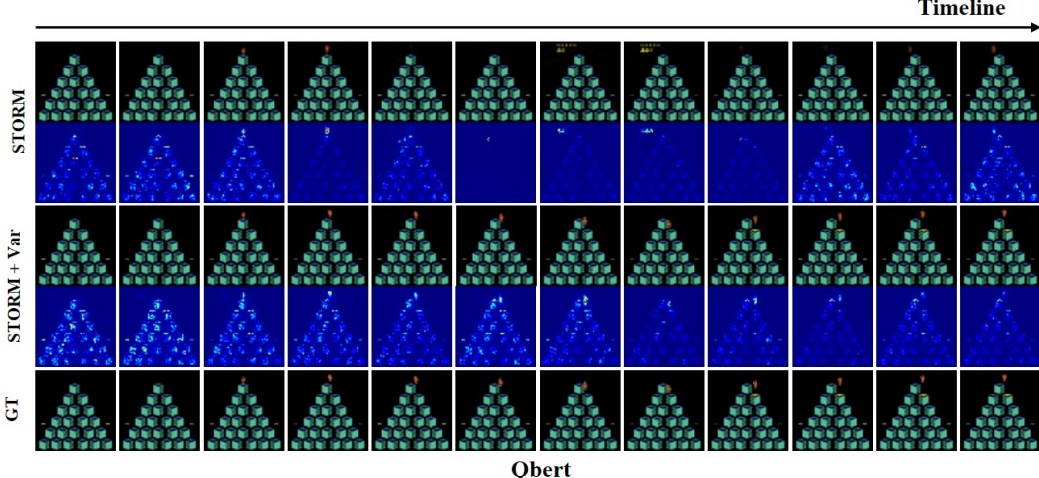

Figure 7: Long-term imagined trajectories from the world model in Qbert games.

## C  PLOT OF DYNAMIC RECONSTRUCTION LOSS WITH VALUE REGULARIZATION

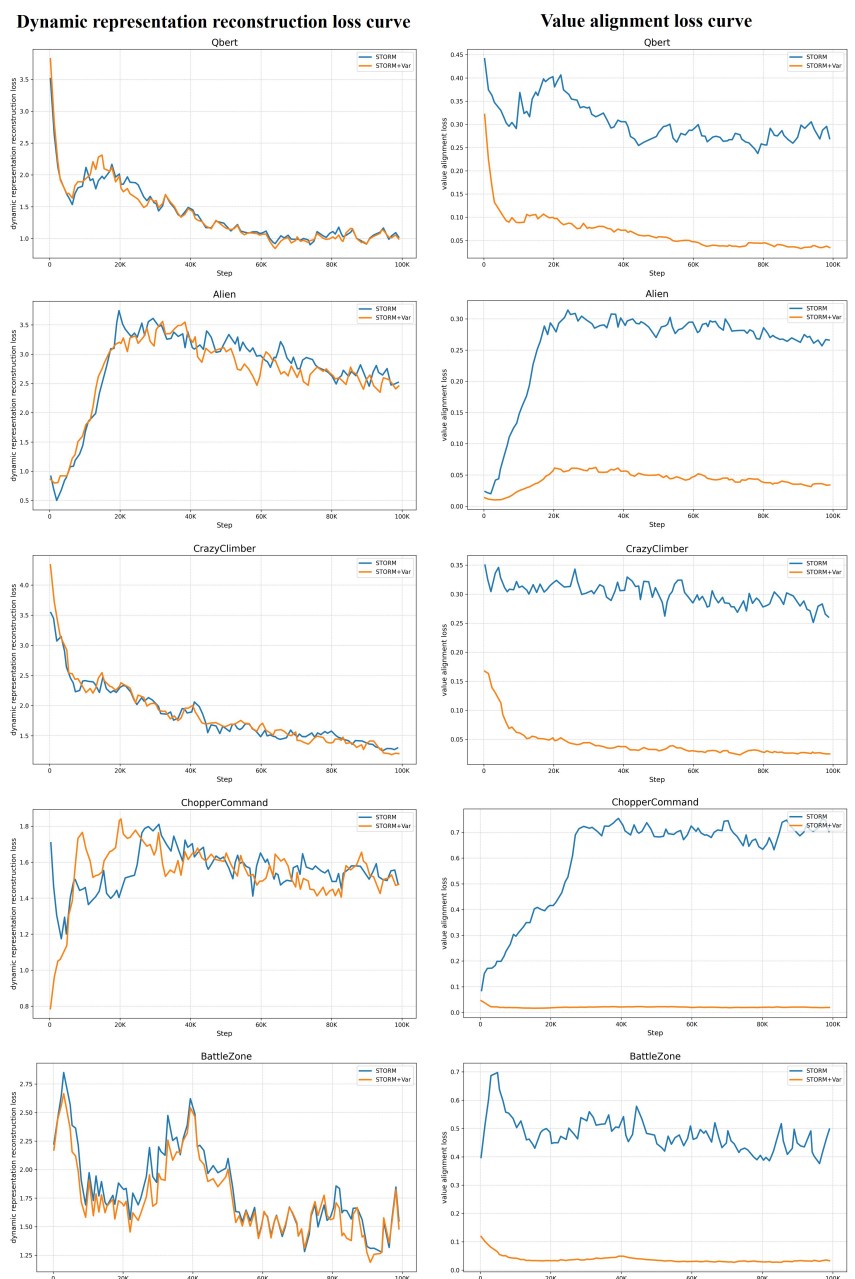

Figure 8: Dynamic reconstruction loss and value alignment loss curve during training.

Figure 8 illustrates the change curves for both the dynamics representation reconstruction loss and the value-alignment loss across different Atari games. The blue curve represents the training trajectory of the original STORM method, while the orange curve represents the training trajectory of our enhanced method, STORM+Var, which incorporates the value-alignment regularization.

The results show that introducing Value-Alignment Regularization maintains the overall dynamics representation reconstruction loss at a level consistent with the baseline, while significantly reducing the value divergence. This result confirms that the performance gain stems from improved value-relevant fidelity within the latent space, achieved by efficiently prioritizing information critical for decision-making without compromising the model's fundamental dynamics reconstruction ability.

# D DISCUSSION ON POTENTIAL VALUE-DELUSIONAL STATES

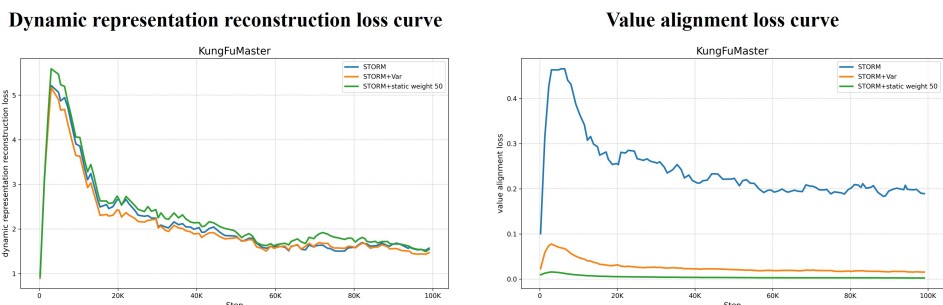

Figure 9: Dynamic reconstruction loss and value alignment loss curve during training.

We investigated the potential problem of the world model generating delusional states that are easy for the value function to predict but are dynamically inconsistent when training excessively favors the value alignment regularization term.

Figure 9 illustrates the change curves of the dynamics representation reconstruction loss and the value alignment loss on the KungFuMaster game. The blue curve represents the training dynamics of the original STORM method; the orange curve shows the dynamics of STORM+Var under adaptive weighting; and the green curve depicts STORM+Var under static weighting (using a large weight, set to 50).

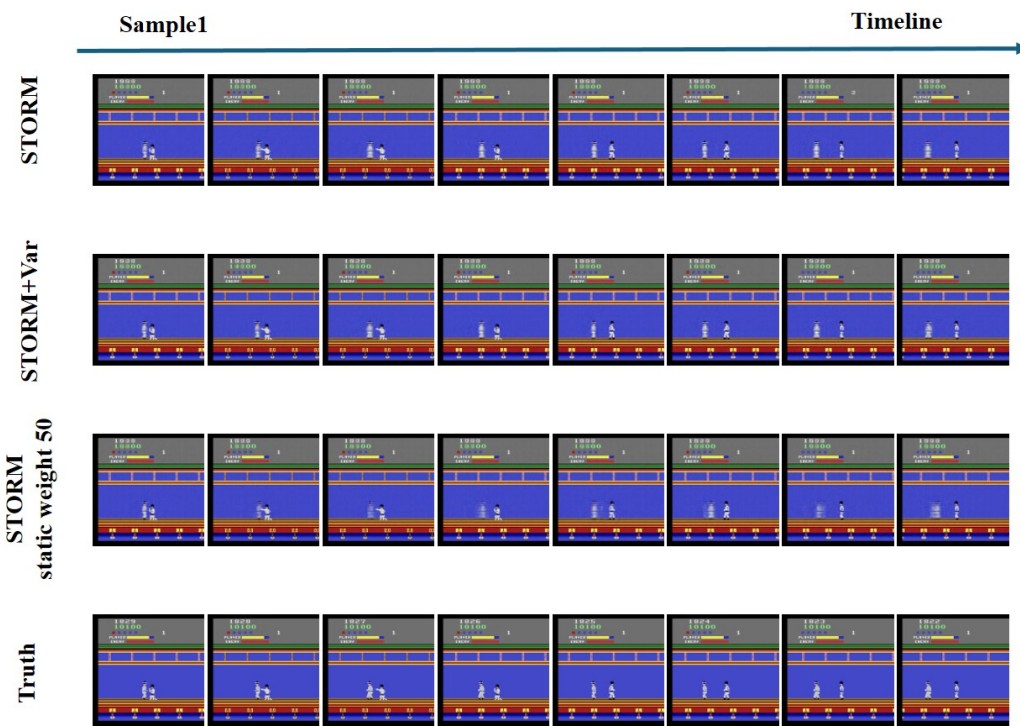

Figure 10: World model visualization result comparisons.

The curve analysis clearly shows that the introduction of value alignment regularization significantly reduces value divergence. However, when training overly favors value alignment, despite achieving a lower value alignment loss, it simultaneously leads to degradation in dynamics reconstruction capability. We further conducted a visualization analysis of the world models trained under these three settings. The results in Visualizations Figure 10 and 11 indicate that when the training is overly

biased toward the value alignment loss, the world model tends to produce delusional states that are easy for the value function to predict but are dynamically inconsistent ( for example, the white enemy's reconstruction exhibits an unstable ghosting artifact in Figure 10, or an enemy suddenly materializes out of thin air in Figure 11). Our designed adaptive weighting mechanism effectively mitigates this issue by dynamically regulating the balance between the dynamics representation reconstruction loss and the value alignment loss.

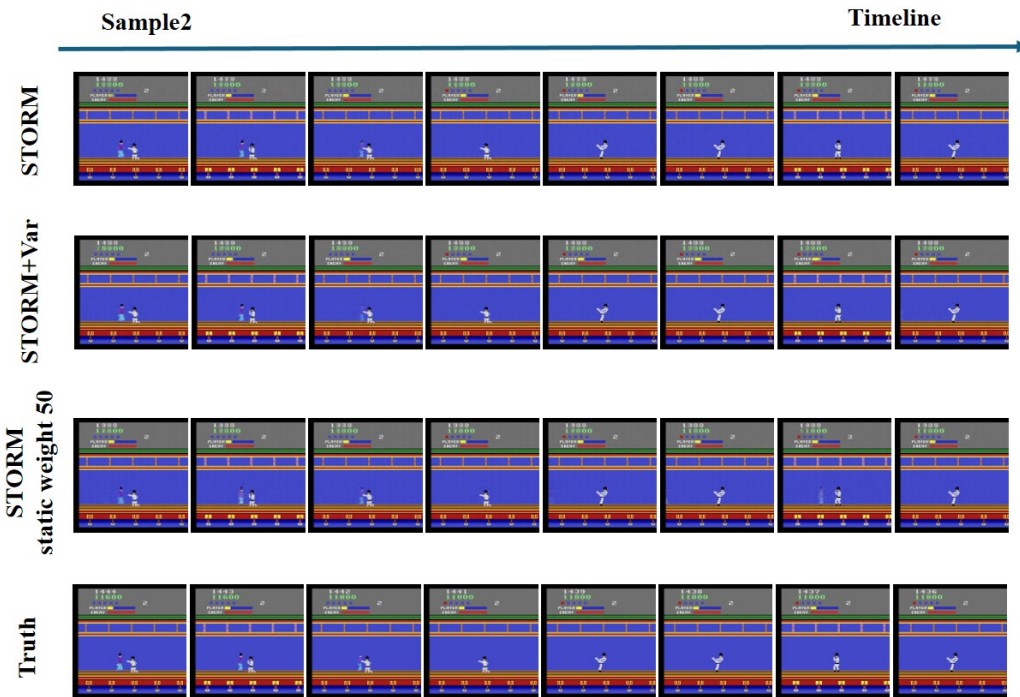

Figure 11: World model visualization result comparisons.

# E   DISCUSSION ON ADAPTIVE WEIGHT $\beta_{var}$

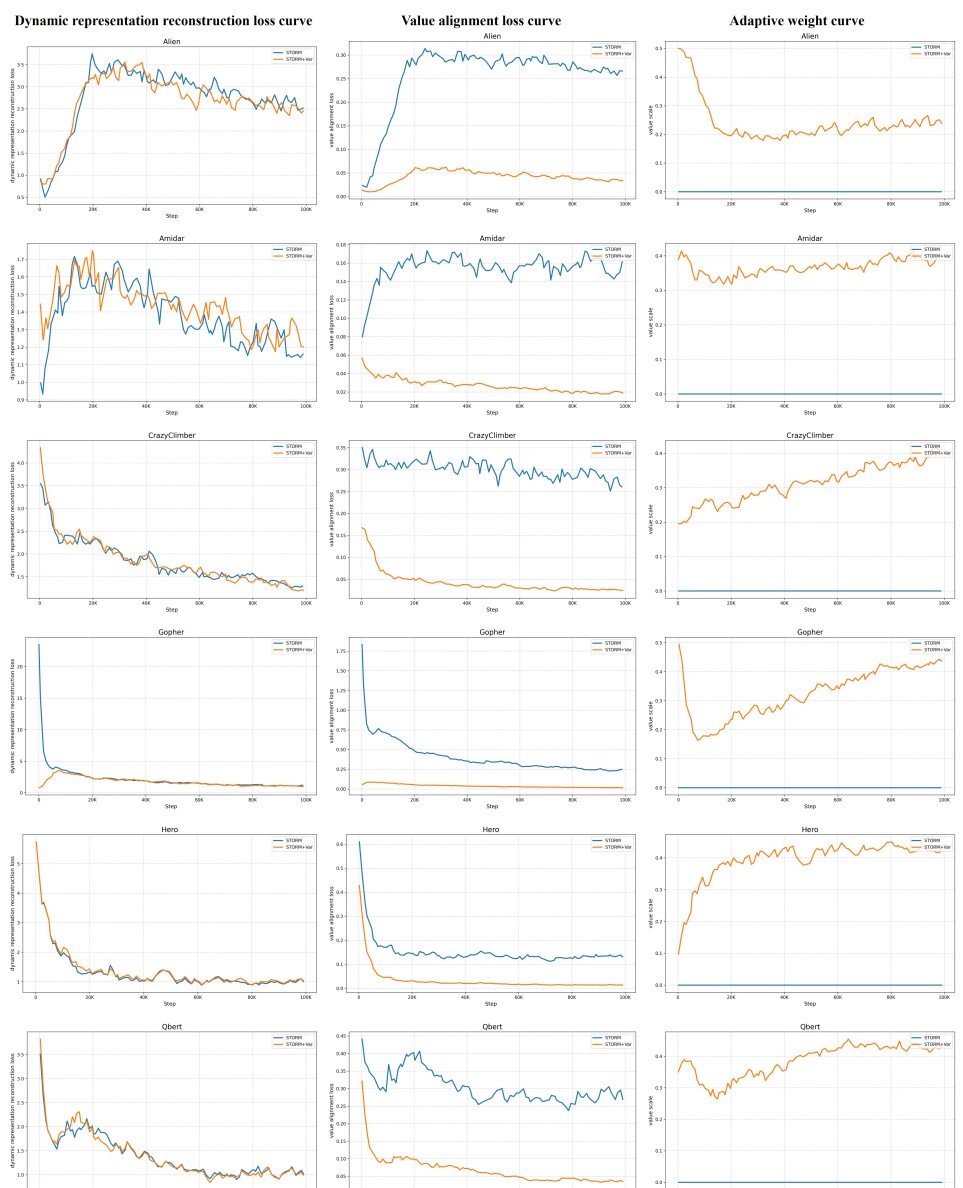

Figure 12: Dynamic reconstruction loss, value alignment loss and adaptive weight curve.

We investigated the trajectory of the adaptive weight during training. Figure 12 illustrates the change curves of the dynamics representation reconstruction loss, the value alignment loss, and the adaptive weight across various Atari games. The blue curve represents the training evolution of the original STORM method, while the orange curve shows the evolution of STORM+Var, which incorporates the value alignment regularization.

The results indicate that in the initial learning stage, the dynamics representation loss is high, and the adaptive weight is decreasing, meaning the primary focus is on Maximum Likelihood Reconstruction. As training stabilizes, the dynamics representation loss decreases, and the adaptive weight increases, signaling a shift in attention toward task-critical features. Crucially, the introduction of adaptive weighting leads to a significant reduction in the value alignment loss without causing an increase in the dynamics representation loss. This demonstrates that maximum likelihood optimization and value alignment are not in antagonism (counteract) but rather achieve synergistic alignment.

# F  DETAILED ANALYSIS OF WORLD MODEL VISUALIZATION RESULTS

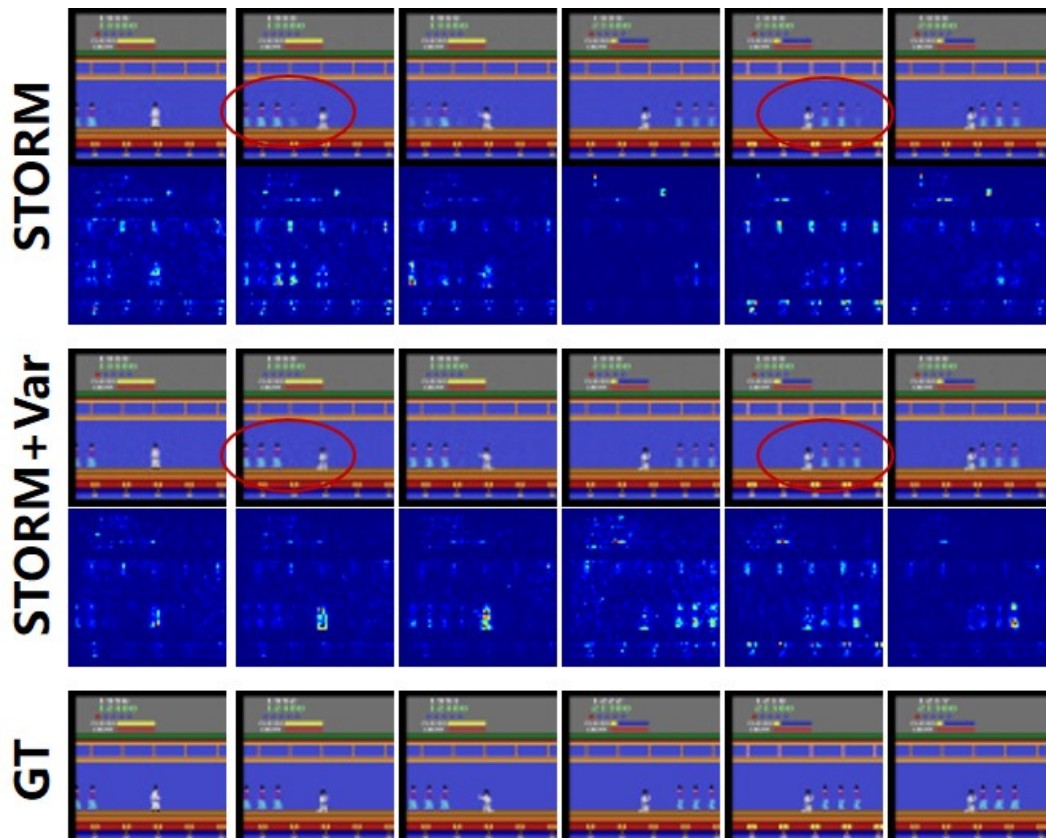

Figure 13: Detailed world model visualization results on KungFuMaster.

Figure 13 presents a comparative visualization of the world models learned by the vanilla STORM and the value-aligned STORM+Var in the KungFuMaster environment. The first row displays the decoded reconstruction results, while the second row illustrates the value-weighted saliency maps of the world model's latent states.

In the KungFuMaster task, the core objective is to control the central white character to defeat enemies approaching from both the left and right sides to clear the stage. Consequently, the critical challenge in this environment lies in accurately reconstructing the agent and the enemies, whereas the static background information can be selectively ignored.

From the visualization results, two key observations can be made: First, regarding the directly decoded reconstructions, the vanilla STORM suffers from incomplete and inaccurate predictions of key task entities (as highlighted by the red circles in Figure 13). The introduction of value alignment effectively resolves this issue. Second, regarding the saliency maps, compared to the vanilla STORM, the incorporation of value alignment enables the world model to focus more intensively on the controlled agent and the approaching enemies, rather than on irrelevant background details.

# G  ORIGINAL REPORTED AND REPRODUCED RESULTS ON THE ATARI 100K.

Table 5: DreamerV3 original and reproduced quantitative results on the Atari 100k benchmark.

| Game | Random | Human | DreamerV3 v1 reported | DreamerV3 v2 reported | DreamerV3 our reproduced | DreamerV3+Var our reproduced |
|---|---|---|---|---|---|---|
| Alien | 227.8 | 7127.7 | 959 | 1118 | 875.88 | 1233.2 |
| Amidar | 5.8 | 1719.5 | 139 | 97 | 143.7 | 185.4 |
| Assault | 222.4 | 742.0 | 706 | 683 | 843.7 | 981.38 |
| Asterix | 210.0 | 8503.3 | 932 | 1062 | 1102.5 | 1162.6 |
| BankHeist | 14.2 | 753.1 | 649 | 398 | 1072.0 | 1121.2 |
| BattleZone | 2360.0 | 37187.7 | 12250 | 20300 | 11138.0 | 12750.0 |
| Boxing | 0.1 | 12.1 | 78 | 82 | 80.3 | 87.4 |
| Breakout | 1.7 | 30.5 | 31 | 10 | 25.3 | 45.6 |
| ChopperCommand | 811.0 | 7387.8 | 420 | 2222 | 1438.0 | 1826.0 |
| CrazyClimber | 10780.5 | 35829.4 | 97190 | 86225 | 89900.0 | 81720.0 |
| DemonAttack | 152.1 | 1971.0 | 303 | 577 | 223.9 | 227.2 |
| Freeway | 0.0 | 29.6 | 0 | 0 | 30.2 | 31.6 |
| Frostbite | 65.2 | 4334.7 | 909 | 3377 | 1628.0 | 347.9 |
| Gopher | 257.6 | 2412.5 | 3730 | 2160 | 1683.9 | 2807.0 |
| Hero | 1027.0 | 30826.4 | 11161 | 13354 | 4994.4 | 9360.6 |
| Jamesbond | 29.0 | 302.8 | 445 | 540 | 332.0 | 542.0 |
| Kangaroo | 52.0 | 3035.0 | 4098 | 2643 | 1529.2 | 3650.4 |
| Krull | 1598.0 | 2665.5 | 7782 | 8171 | 8364.8 | 9821.4 |
| KungFuMaster | 258.5 | 22736.3 | 21420 | 25900 | 16375.0 | 21075.0 |
| MsPacman | 307.3 | 6951.6 | 1327 | 1521 | 1947.0 | 1749.5 |
| Pong | -20.7 | 14.6 | 18 | -4 | 19.1 | 19.8 |
| PrivateEye | 24.9 | 69571.3 | 882 | 3238 | 2331.2 | -115.6 |
| Qbert | 163.9 | 13455 | 3405 | 2921 | 1223.5 | 2267.8 |
| RoadRunner | 11.5 | 7845.0 | 15565 | 19230 | 9868.6 | 14704.0 |
| Seaquest | 68.4 | 42054.7 | 618 | 962 | 513.2 | 546.3 |
| UpNDown | 533.4 | 11693.2 | 7567 | 46910 | 12679.2 | 18485.4 |
| Mean ($\uparrow$) | 0.00 | 1.00 | 1.12 | 1.25 | 1.10 | 1.34 |
| Median ($\uparrow$) | 0.00 | 1.00 | 0.49 | 0.49 | 0.58 | 1.00 |

Table 5 presents the original recorded results for both the v1 and v2 versions of DreamerV3, alongside our reproduced results using the PyTorch implementation, on the Atari 100K dataset.

Table 6: STORM original and reproduced quantitative results on the Atari 100k benchmark.

| Game | Random | Human | STORM ori reported | STORM reproduced by Meo et al. (2024) | STORM our reproduced | STORM+Var our reproduced |
|---|---|---|---|---|---|---|
| Alien | 227.8 | 7127.7 | 984 | 1364 | 1054.3 | 1361.4 |
| Amidar | 5.8 | 1719.5 | 205 | 239 | 177.29 | 248.36 |
| Assault | 222.4 | 742.0 | 801 | 707 | 715.9 | 752.55 |
| Asterix | 210.0 | 8503.3 | 1028 | 865 | 1276.0 | 1535.0 |
| BankHeist | 14.2 | 753.1 | 641 | 375 | 1060.5 | 935.0 |
| BattleZone | 2360.0 | 37187.7 | 13540 | 10780 | 7080.0 | 10140.0 |
| Boxing | 0.1 | 12.1 | 80 | 80 | 78.6 | 83.0 |
| Breakout | 1.7 | 30.5 | 16 | 12 | 20.88 | 26.43 |
| ChopperCommand | 811.0 | 7387.8 | 1888 | 2293 | 1768.0 | 1695.0 |
| CrazyClimber | 10780.5 | 35829.4 | 66776 | 54707 | 47473.0 | 57335.0 |
| DemonAttack | 152.1 | 1971.0 | 165 | 229 | 194.6 | 204.6 |
| Freeway | 0.0 | 29.6 | 33.5 | 0 | 29.7 | 32.0 |
| Frostbite | 65.2 | 4334.7 | 1316 | 646 | 258.8 | 260.2 |
| Gopher | 257.6 | 2412.5 | 8240 | 2631 | 8551.0 | 13509.6 |
| Hero | 1027.0 | 30826.4 | 11044 | 11044 | 12249.2 | 12574.0 |
| Jamesbond | 29.0 | 302.8 | 509 | 552 | 446.4 | 462.5 |
| Kangaroo | 52.0 | 3035.0 | 4208 | 1716 | 1542.0 | 3322.6 |
| Krull | 1598.0 | 2665.5 | 8412 | 6869 | 8360.1 | 8896.5 |
| KungFuMaster | 258.5 | 22736.3 | 26182 | 20144 | 15760 | 26615.0 |
| MsPacman | 307.3 | 6951.6 | 2674 | 2673 | 1906.9 | 2417.3 |
| Pong | -20.7 | 14.6 | 11 | 8 | 20.6 | 20.2 |
| PrivateEye | 24.9 | 69571.3 | 7781 | 2734 | 414.4 | 2584.7 |
| Qbert | 163.9 | 13455 | 4523 | 2986 | 2912.5 | 4243.4 |
| RoadRunner | 11.5 | 7845.0 | 17564 | 12477 | 11523.0 | 13999.0 |
| Seaquest | 68.4 | 42054.7 | 525 | 525 | 441.4 | 430.0 |
| UpNDown | 533.4 | 11693.2 | 7985 | 7985 | 6406.4 | 8982.6 |
| Mean (↑) | 0.00 | 1.00 | 1.27 | 0.95 | 1.14 | 1.36 |
| Median (↑) | 0.00 | 1.00 | 0.58 | 0.36 | 0.51 | 0.81 |

Table 6 presents the original recorded results for STORM, third-partyMeo et al. (2024) reproduction results, and our reproduced results using the official STORM codebase on the Atari 100K dataset.

# H  MORE ABLATION EXPERIMENTS

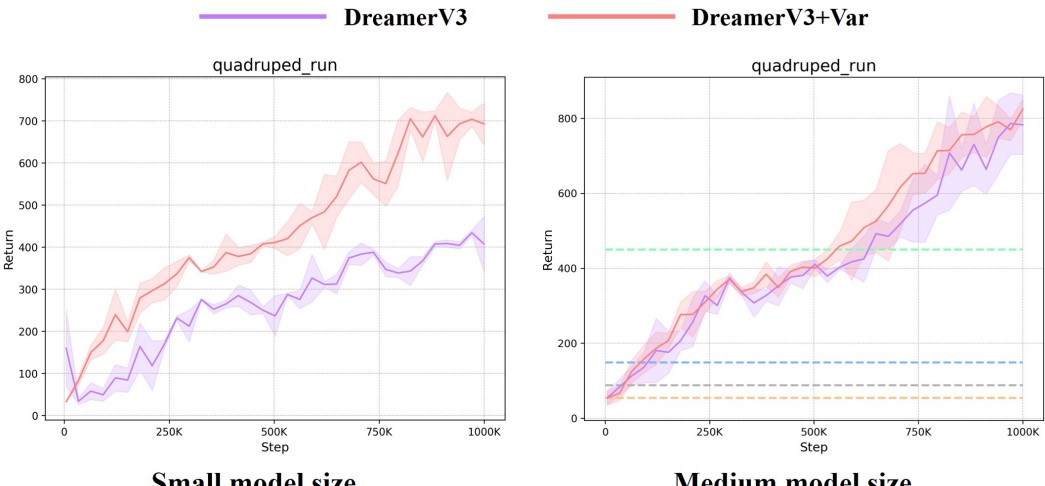

**Small model size**    **Medium model size**

Figure 14: Results on DMC Quadruped run with different model sizes.

In order to further validate the potential of our proposed Value Alignment Regularization under model capacity constraints, we conducted tests on the DeepMind Control Suite's quadruped run environment. Unlike the Atari 100K setting which typically uses 100K training steps, we increased the training steps to 1 Million for quadruped run to ensure model convergence.

Figure 14 illustrates the training curves for both the small and medium model sizes. Under the medium model size, the introduction of value alignment regularization yields performance comparable to the original DreamerV3 algorithm. However, when the model size is reduced, the convergence performance of the vanilla DreamerV3 algorithm degrades significantly on the quadruped run environment. Crucially, introducing value alignment at this reduced capacity effectively mitigates this performance drop, thereby raising the performance ceiling of the model under constraints.

# I  DISCUSSION ON FAILURE ENVIRONMENTS

We hypothesize that the failure in the PrivateEye and Frostbite environments for DreamerV3+Var stems from their nature as notoriously sparse-reward environments requiring significant initial exploration. During the entire phases of training, the value targets are inherently noisy and non-informative. Forcing the world model to align with such a noisy value function via the $L_{var}$ term may induce "overfitting," which drastically reduces the variance in the latent space that is crucial for effective exploration. Unlike Maximum Likelihood Estimation, which naturally encourages capturing all diversity in the observations, premature value alignment in sparse-reward settings can inadvertently hinder the discovery of the first critical reward signal.

# J  CLARIFICATION OF THE DEFINITION OF UPPER AND LOWER BOUNDS

The concepts of lower and upper bounds are better understood as conceptual metaphors for the model's performance capabilities. Specifically, the lower bound refers to the model's fidelity to the simulated world: employing the RSSM/TSSM architecture to learn dynamic representations via Maximum Likelihood Estimation (MLE) and reconstructing visual inputs guarantees that the learned representations are at least physically consistent, thereby preventing hallucinations that completely deviate from the environment. Conversely, the upper bound refers to the model's utility for a specific task: under finite capacity, the MLE approach tends to focus on irrelevant backgrounds, potentially neglecting task-critical features; introducing value alignment guidance rectifies this, consequently raising the ceiling of task performance.

# K  CODE AND DECLARATIONS

We implement our method on the Torch-version DreamerV3 code and official STORM code. For detailed code implementation, please refer to the supplementary materials.In the Freeway environment of Atari 100k, we applied the same trick as used in IRIS(Micheli et al., 2022).

In our work, the large language model is used solely for text refinement and grammar correction, with no other applications.

# L  DETAILED MODEL STRUCTURE AND HYPERPARAMETER

## L.1  STORM

The network architecture and parameters for the STORM model are consistent with those in (Zhang et al., 2023). The specific architecture and parameters are detailed below.

Table 7: Image Encoder Architecture and Parameters: The image encoder takes an input image of size $3 \times 64 \times 64$ and consists of four convolutional blocks, followed by Flatten, Linear, and Reshape layers. Each convolutional block is composed of a Conv layer, a BN layer, and a ReLU activation function. The Conv layer (LeCun et al., 1989) has a kernel size of 4, a stride of 2, and a padding of 1. The BN layer (Ioffe & Szegedy, 2015) is used for batch normalization. The Flatten and Reshape layers are used to adjust the tensor indexing.

| Module | Output Tensor Shape |
| --- | --- |
| Input: Environment Image ($o_t$) | $3 \times 64 \times 64$ |
| Convolutional Block 1 (Conv + BN + ReLU) | $32 \times 32 \times 32$ |
| Convolutional Block 2 (Conv + BN + ReLU) | $64 \times 16 \times 16$ |
| Convolutional Block 3 (Conv + BN + ReLU) | $128 \times 8 \times 8$ |
| Convolutional Block 4 (Conv + BN + ReLU) | $256 \times 4 \times 4$ |
| Flatten | 4096 |
| Linear | 1024 |
| Reshape | $32 \times 32$ |
| Output: distribution ($\mathcal{Z}_t$) | $32 \times 32$ |

Table 8: Image Decoder Architecture and Parameters: The image decoder takes a $32 \times 32$ sampled value, $z_t$, as input. The network architecture consists of DeConv modules, which are composed of a DeConv layer Zeiler et al. (2010), a BN layer, and a ReLU activation function. The DeConv layers have a kernel size of 4, a stride of 2, and a padding of 1.

| Module | Output Tensor Shape |
| --- | --- |
| Input: Random Sample ($z_t$) | $32 \times 32$ |
| Flatten | 1024 |
| Linear + BN + ReLU | 4096 |
| Reshape | $256 \times 4 \times 4$ |
| DeConv Block 1 (DeConv + BN + ReLU) | $128 \times 8 \times 8$ |
| DeConv Block 2 (DeConv + BN + ReLU) | $64 \times 16 \times 16$ |
| DeConv Block 3 (DeConv + BN + ReLU) | $32 \times 32 \times 32$ |
| DeConv | $3 \times 64 \times 64$ |
| Output: Decoded Image ($\hat{o}_t$) | $3 \times 64 \times 64$ |

Table 9: Action Mixer Architecture and Parameters: The Action mixer takes a $32 \times 32$ sampled value, $z_t$, and a A-dimensional action as input (where the action dimension varies from 3 to 18 depending on the game). Concatenate merges the last dimension of the two tensors. D is the feature dimension of the Transformer. LN denotes layer normalization Ba et al. (2016).

| Module | Output Tensor Shape |
|---|---|
| Input: Random Sample ($z_t$), Action ($a_t$) | $32 \times 32$, A |
| Reshape and concatenate | 1024 + A |
| Linear + LN + ReLU | D |
| Linear2 + LN2 | D |
| Output: $e_t$ | D |

Table 10: Positional Encoding Module: The Positional Encoding Module adds a learnable parameter matrix, $w_{1:T}$, to the input tensor, $e_{1:T}$. The operation is represented as $e_{1:T} + w_{1:T}$, where the sequence length is denoted by $T$ and the feature dimension by $D$. The matrix $w_{1:T}$ has a shape of $T \times D$. Following the addition, Layer Normalization (LN) is applied.

| Module | Output Tensor Shape |
|---|---|
| Input: $e_{1:T}$ | $T \times D$ |
| Add + LN | $T \times D$ |
| Output: $x$ | $T \times D$ |

Table 11: Transformer Module

| Module | Sub-Module | Output Tensor Shape |
|---|---|---|
| Input | $x$ | $T \times D$ |
| MHSA | Multi-head self attention | $T \times D$ |
| | Linear + Dropout | $T \times D$ |
| | Residual | $T \times D$ |
| | LN | $T \times D$ |
| FFN | Linear + ReLU | $T \times 2D$ |
| | Linear + Dropout | $T \times D$ |
| | Residual | $T \times D$ |
| | LN | $T \times D$ |
| Output: | $h_{1:T}$ | $T \times D$ |

Table 12: Transformer-Based Sequence Model Architecture and Parameters: The Positional encoding module is defined in the Table 10 for Positional encoding module. The Transformer block module is defined in the Table 11 for Transformer module.

| Module | Output Tensor Shape |
|---|---|
| Input: $e_{1:T}$ | $T \times D$ |
| Positional encoding | $T \times D$ |
| Transformer blocks $\times$K | $T \times D$ |
| Output: $h_{1:T}$ | $T \times D$ |

Table 13: Other MLP Modules: This table details the architecture of other pure MLP modules.

| Module | Number of MLP Layers | Input Dim | Hidden Dim | Output Dim |
|---|---|---|---|---|
| Dynamics head $g_\phi^D$ | 1 | D | / | 1024 |
| Reward predictor $g_\phi^R$ | 3 | D | D | 255 |
| Continuation predictor $g_\phi^C$ | 3 | D | D | 1 |
| Policy network $\pi_\theta(a_t|s_t)$ | 3 | D | D | A |
| Critic network $V\psi(s_t)$ | 3 | D | D | 255 |

Table 14: STORM Network Hyperparameters.

| Hyperparameter | Symbol | Value |
|---|---|---|
| Transformer layers | $K$ | 2 |
| Transformer feature dimension | $D$ | 512 |
| Transformer heads | – | 8 |
| Dropout probability | $P$ | 0.1 |
| World model training batch size | $B_1$ | 16 |
| World model training batch length | $T$ | 64 |
| Imagination batch size | $B_2$ | 1024 |
| Imagination context length | $C$ | 8 |
| Imagination horizon | $L$ | 16 |
| Update world model every env step | – | 1 |
| Update agent every env step | – | 1 |
| Environment context length | – | 16 |
| Gamma | $\gamma$ | 0.985 |
| Lambda | $\lambda$ | 0.95 |
| Entropy coefficient | $\eta$ | $3 \times 10^{-4}$ |
| Critic EMA decay | $\sigma$ | 0.98 |
| Optimizer | – | Adam |
| World model learning rate | – | $1.0 \times 10^{-4}$ |
| World model gradient clipping | – | 1000 |
| Actor-critic learning rate | – | $3.0 \times 10^{-5}$ |
| Actor-critic gradient clipping | – | 100 |
| Gray scale input | – | False |
| Frame stacking | – | False |
| Frame skipping | – | 4 |
| Use of life information | – | True |

## L.2 DREAMERV3

The DreamerV3 network architecture and parameters remain consistent with those detailed in (Hafner et al., 2025). Table15 are the hyperparameters.

Table 15: DreamerV3 Network Hyperparameters

| Hyperparameter | Value |
|---|---|
| Replay capacity | $5 \times 10^6$ |
| Batch size | 16 |
| Batch length | 64 |
| Activation | RMSNorm+SiLU |
| Learning rate | $4 \times 10^{-5}$ |
| Gradient clipping | AGC(0.3) |
| Optimizer | LaProp($\epsilon = 10^{-20}$) |
| World Model reconstruction loss scale | 1 |
| World Model dynamics loss scale | 1 |
| World Model representation loss scale | 0.1 |
| World Model latent unimix | 1% |
| World Model free nats | 1 |
| Actor-Critic imagination horizon | 15 |
| Actor-Critic return lambda | 0.95 |
| Critic loss scale | 1 |
| Critic replay loss scale | 0.3 |
| Critic EMA regularizer | 1 |
| Critic EMA decay | 0.98 |
| Actor loss scale | 1 |
| Actor entropy regularizer | $1 \times 10^{-3}$ |
| Actor unimix | 1% |
| Actor RetNorm scale | $\text{Per}(R, 95) - \text{Per}(R, 5)$ |
| Actor RetNorm limit | 1 |
| Actor RetNorm decay | 0.99 |

