# OpenReview forum: "Value-aligned World Model Regularization for Model-based Reinforcement Learning"
_ICLR.cc/2026/Conference — ICLR 2026 Conference Withdrawn Submission_

### Official Review · Reviewer_MsDe · 2025-10-26

**Soundness:** 3
**Presentation:** 3
**Contribution:** 3
**Rating:** 4
**Confidence:** 4

**Summary:**

This paper explores incorporating a value regularization loss into world model learning.
Value regularization encourages the learned model to align predicted states with their corresponding value estimates, helping it focus on task-relevant features.
Albeit simple, it demonstrates substantial improvement on the Atari 100k benchmark (and slight improvement on DMC) when applied to DreamerV3 and STORM.

**Strengths:**

**S1. Simplicity of the method**
* The proposed modification is conceptually simple and easy to implement; it adds a KL-based regularization term on the value during model learning (after a warm-up phase).
* The method can be seamlessly integrated into existing model-based RL frameworks (such as DreamerV3 and STORM).

**S2. Substantial improvement and interpretability**
* Despite its simplicity, the proposed method yields significant improvements in the Atari 100k benchmark (e.g., HNS mean: 1.1 -> 1.34 for DreamerV3 and 1.14 -> 1.36 for STORM).
* Visualization results show that the world model focuses more on task-relevant objects, supporting the claim that value regularization enhances the quality of the representation.

**Weaknesses:**

**W1. Experimental results for the baselines are different (worse) from the original paper**
* In the paper, DreamerV3 and STORM scores 1.10 and 1.14, while the score in the original paper is 1.25 and 1.27 (HNS mean in Atari 100k); DreamerV3 scores 817, 827, while the score in the original paper is 871, 861 (Task mean).
* While authors mentioned that they used torch version of DreamerV3, but I wonder the implementation correctly reproduces DreamerV3, as there is substantial gap in the performance (I tried to take a look on the code, but I couldn’t find the DreamerV3 code in the supplementary materials).
* Moreover, while I assume that STORM is ran with the official implementation (as the paper builds on top of official codebase of STORM), but still shows a large discrepancy.

**Questions:**

**Q1. Clarification on the results**
* Why the numbers are different from the original DreamerV3 and STORM paper?  Could the authors clarify whether differences in architecture, hyperparameters, or experimental setup could explain this discrepancy?
* It would also help if the DreamerV3 code used for experiments were made available in the supplementary materials.

**Q2. Experiments with distracting dimensions**
* For DMC, all state dimensions are task-relevant, so value regularization might not strongly affect performance.
* How about adding distracting or irrelevant dimensions (as in [1]) to test whether the proposed method helps the model focus on the necessary states?
* This could provide stronger evidence for the intended mechanism.

**Q3. Plot of reconstruction loss with value regularization**
* I think it would be interesting to analyze how value regularization influences the reconstruction loss.
* Specifically, does the improvement come from 1) higher overall reconstruction loss but better value-relevant fidelity, or 2) lower overall reconstruction loss due to improved learning signals?
* Although not critical, such an analysis would clarify how value regularization benefits model learning.

PLEASE READ: I would like to emphasize that I believe this paper is strong and has substantial potential.
However, **Q1 (W1) is a critical concern** for me. I currently lean toward a weak reject, but if the authors can provide the clarification, I'm inclined to significantly raise my recommendation.

---

> ### Author Response · Authors · 2025-11-25
>
> Dear Reviewer MsDe,
>
> We sincerely appreciate your positive comments and constructive suggestions, particularly your attention to the experimental baselines. We have prepared detailed clarifications and additional experimental evidence to address your questions below:
>
> ---
>
> **Q1: Why the numbers are different from the original DreamerV3 and STORM paper?  Could the authors clarify whether differences in architecture, hyperparameters, or experimental setup could explain this discrepancy?**
>
> **A1:** We sincerely thank the reviewer for this critical observation. We have carefully re-examined the literature and identified the source of this discrepancy, which stems from specific version configurations used in the original DreamerV3 paper.
>
> **1. DreamerV3 Version Differences (arXiv v1 vs. v2):** There are two distinct versions of the DreamerV3 paper available on arXiv, which share the same architecture but differ significantly in model scale and hyperparameters for the Atari 100k benchmark:
>
> -  arXiv v1 (https://arxiv.org/abs/2301.04104v1): Utilizes a smaller model (~18M parameters) and a Training Ratio of 1024.
>
> - arXiv v2 (https://arxiv.org/pdf/2301.04104v2): Upgrades to a significantly larger model (~200M parameters) and adjusts the Training Ratio to 128.
>
> **2. Consistency of Our Reproduction:** Our experiments utilized the widely adopted PyTorch implementation (https://github.com/NM512/dreamerv3-torch), which adheres to the specifications of the v1 paper. As shown in the tables above, our reproduced baseline results (e.g., Atari Mean 1.10) align closely with the v1 reported figures (1.12). The higher scores cited by the reviewer (1.25) correspond to the v2 setting, which benefits from a 10x larger parameter count.
>
> **3. Alignment with Community Standards:** We further verified that this benchmarking standard is consistent with other recent publications. For instance, state-of-the-art methods like STORM and TWISTER also benchmark against the v1 version of DreamerV3.
>
> Comparison of Reported vs. Reproduced Scores (The specific scores for each sub-environment can be found in the newly added **Appendix G, Tables 5 and 6**.):
>
> | Atari 100K      | DreamerV3 v1 reported |DreamerV3 v2 reported | DreamerV3 our reproduced | DreamerV3+Var our reproduced |
> | :---        |    :----:   |    :----:   |  :----:   |        ---: |
> | Mean      | 1.12       | 1.25  | 1.10       | 1.34  |
> | Median   | 0.49        | 0.49      | 0.58       | 1.00  |
>
> | DMC Vision  | DreamerV3 v1 reported |DreamerV3 v2 reported | DreamerV3 our reproduced | DreamerV3+Var our reproduced |
> | :---        |    :----:   |    :----:   |  :----:   |        ---: |
> | Mean      | 740       | 786  | 792       | 827  |
> | Median   | 809      | 861     | 877       | 894  |
>
> | DMC Proprioceptive  | DreamerV3 v1 reported |DreamerV3 v2 reported | DreamerV3 our reproduced | DreamerV3+Var our reproduced |
> | :---        |    :----:   |    :----:   |  :----:   |        ---: |
> | Mean      | 743       | 754  | 805       | 817  |
> | Median   | 846      | 871     | 881       | 901  |
>
>
> ---

---

> ### Author Response · Authors · 2025-11-25
>
> ---
>
> Regarding the STORM baseline, we confirm that our experiments were conducted using the official STORM codebase. To further validate our results, we cross-referenced other third-party reproductions (The specific scores for each sub-environment can be found in the newly added **Appendix G, Tables 5 and 6**.):
>
> | Atari 100K      | STORM original reported |STORM reproduced by [1]  | STORM our reproduced | STORM+Var our reproduced |
> | :---        |    :----:   |    :----:   |  :----:   |        ---: |
> | Mean      | 1.27       | 0.95  | 1.14       | 1.36  |
> | Median   | 0.58        | 0.36      | 0.51       | 0.81  |
>
> **1. Alignment with Third-Party Reproductions:** The discrepancy between "Original Paper Scores" and "Third-Party Reproductions" is a known phenomenon in RL, often stemming from differences in evaluation protocols (e.g., hardware determinism, seed variance). Notably, ICLR 2025 [1] reports a reproduced STORM mean score of 0.95 on Atari 100k, indicating that the original paper's scores are often difficult to match due to specific evaluation setups. Our reproduction yields a mean of 1.14, which is higher than this third-party benchmark, confirming the robustness of our baseline implementation.
>
> **2. Empirical Analysis of Seed Sensitivity:** To explicitly demonstrate the impact of stochasticity, we conducted additional experiments with more random seeds on the specific sub-environments (e.g., BattleZone, CrazyClimber) where the discrepancy between our reproduced results and the original STORM paper was most pronounced. The results are detailed in the table below:
>
> | seed | 100 | 102  | 103 | 104 | 105 | 106  | 107 | 108 | 109 | 110 |
> | :---        |    :----:   |    :----:   |  :----:   |  :----:   |    :----:   |  :----:   | :----:   |    :----:   |  :----:   |       ---: |
> | results on BattleZone   | 6500 | 9900 | 5000 | 4250 | 9750  | 12650 |  16450 | 5100 | 2350 | 8500 |
> | results on CrazyClimber  | 52300 | 41455 | 42920 | 49415 |51275  | 53620  |  47835 | 62825 | 53890 | 52385 |
>
> As shown, performance varies drastically across seeds (e.g., 2350 vs. 16450 in BattleZone). This high variance largely explains the gap between our reported means and those in the original paper.
>
> Ultimately, while variations in model scale, hyperparameters, hardware, and random seeds can shift absolute scores, internal validity remains paramount. Most importantly, we ensured a strict **"apples-to-apples"** comparison: both the baselines (DreamerV3/STORM) and our method (Var) were trained using identical codebases, hyperparameters, random seeds, and compute environments. Therefore, the substantial relative improvements we report are robust and statistically significant, regardless of the absolute baseline floor.
>
> [1] Masked Generative Priors Improve World Models Sequence Modelling Capabilities, ICLR 2025.
>
> ---

---

> ### Author Response · Authors · 2025-11-25
>
> ---
>
> **Q2: It would also help if the DreamerV3 code used for experiments were made available in the supplementary materials.**
>
> **A2:** We have added the DreamerV3 code used in our experiments to the latest supplementary materials. We warmly welcome the reviewer to review and test it to verify our implementation.
>
> ---
>
> **Q3: How about adding distracting or irrelevant dimensions to test whether the proposed method helps the model focus on the necessary states?**
>
> **A3:** We greatly appreciate this insightful suggestion. To empirically verify whether our method helps the model focus on task-relevant states, we conducted an additional experiment on the DMC suite by systematically injecting random, irrelevant feature dimensions into the input observations at varying ratios.
>
> | Noise Ratio      | Insert 0% noise  | Insert 10% noise | Insert 20% noise | Insert 50% noise |
> | :---        |    :----:   |    :----:   |  :----:   |        ---: |
> | DreamerV3 results on  Hopper Hop   | 236       | 224  | 197      | 134  |
> | DreamerV3+Var results on  Hopper Hop   | 238       | 231  | 219       | 187  |
>
> As shown in the table, the performance of the baseline DreamerV3 degrades noticeably as the ratio of irrelevant noise features increases. In contrast, Value-Alignment Regularization effectively mitigates this degradation, maintaining robust performance as noise levels rise.
>
> ---
>
> **Q4: Specifically, does the improvement come from 1) higher overall reconstruction loss but better value-relevant fidelity, or 2) lower overall reconstruction loss due to improved learning signals?**
>
> **A4:** This is a insightful question regarding the underlying optimization dynamics. We have included the training curves for both the Value-Alignment Loss and the Dynamics Representation Reconstruction Loss in the newly added **Appendix C, Figure 8**.
>
> The results show that introducing value-alignment regularization maintains the overall dynamics representation reconstruction loss at a level consistent with the baseline, rather than increasing or decreasing it significantly. Simultaneously, it significantly reduces the value divergence.
>
> This confirms that the performance gain stems from improved value-relevant fidelity within the latent space. The model achieves this by efficiently prioritizing information critical for decision-making without compromising its general ability to reconstruct environmental dynamics.

---

> > ### Comment · Reviewer_MsDe · 2025-11-26
> >
> > Thank you for the clarification, providing the DreamerV3 implementation, and the additional experiments.
> >
> > I believe that this work will be a great contribution in model-based RL community, as it assesses the problem of learning a better world model with a generally applicable solution (i.e., without additional data or supervision).
> >
> > As all my concern is resolved, I increased the score from 4 to 8.

---

### Official Review · Reviewer_Hona · 2025-10-27

**Soundness:** 3
**Presentation:** 3
**Contribution:** 3
**Rating:** 4
**Confidence:** 3

**Summary:**

This paper proposes a method to integrate value-alignment into world model training for model-based RL. The key idea is adding a value-alignment regularization term (Var) to the world model loss, using a KL divergence between the value function's distribution on real and imagined states. The authors argue this combines the stability of maximum likelihood methods (lower bound) with the decision-awareness of value-aware methods (upper bound), and introduce a warmup phase and an adaptive weighting mechanism to stabilize the training. The idea is evaluated on Atari100k and DMC, showing consistent improvements when added to DreamerV3 and STORM.

**Strengths:**

+ the integration of value-alignment into world model training as a KL divergence term on the value network's distributional output is a simple, elegant, and easy-to-implement idea
+ the claim and validation of "RSSM/TSSM ensures the lower bound... while value awareness enhances the upper bound" offers new insights and understanding
+ validation across visual and proprioceptive as well as discrete and continuous control tasks is strong

**Weaknesses:**

- The exact definition of lower bound and upper bound is missing and makes the claim vague.
- the proposed method should have been compared against modern value-aware methods such as VaGraM or CVAML across all tasks
- the failures of the proposed method on e.g., PrivateEye for DreamerV3+Var, Frostbite are not analyzed

**Questions:**

1) How does the method conceptually and empirically differ from prior value gradient weighting methods like VaGraM, beyond the obvious architectural differences (RSSM/TSSM vs RNN)?
2) The lower/upper bounds concept is central to your motivation. Can you provide a more formal definition or a toy example that illustrates this concept?
3) Have you tried a direct, large-scale comparison? The proposed method regularizes the world model to produce states yielding a similar distribution to real states. Will it lead to delusional states that are easy for the value function to predict but are world model dynamically inconsistent?

---

> ### Author Response · Authors · 2025-11-25
>
> Dear Reviewer Hona,
>
> We sincerely appreciate your positive comments and constructive suggestions, which have significantly strengthened our manuscript. In response to your questions, we have prepared the following detailed answers:
>
> ---
>
> **Q1: The exact definition of lower bound and upper bound is missing and makes the claim vague. Can you provide a more formal definition or a toy example that illustrates this concept?**
>
> **A1:** We thank the reviewer for this constructive suggestion! In newly added **Appendix J**, we have clarified that the concepts of "lower" and "upper" bounds serve as conceptual metaphors for the model's performance capabilities:
>
> - **Lower Bound (Fidelity & Validity):** This refers to the model's fidelity to the physical simulation. By employing the RSSM/TSSM architecture to learn dynamic representations guarantees that the learned representations are at least physically consistent. This prevents the model from collapsing into hallucinations that completely deviate from the ground-truth environment dynamics.
>
> - **Upper Bound (Utility & Optimality):** This refers to the model's utility for a specific task. Under finite capacity constraints, a pure MLE approach often wastes capacity on high-entropy but task-irrelevant backgrounds. Introducing Value-Alignment regularization acts as an inductive bias that rectifies this focus, thereby raising the performance ceiling (upper bound) by ensuring the representation captures task-critical features.
>
> ---
>
> **Q2: the proposed method should have been compared against modern value-aware methods such as VaGraM or CVAML across all tasks.**
>
> **A2:** We appreciate this suggestion. To address this, we are currently conducting comparative experiments with VaGraM across the 26 Atari environments. Due to time and resource limitations, we have only completed training on a subset of these environments so far, and the experimental results are summarized in the table below:
>
> | Atari 100K    | Alien | Amidar | BankHeist | BattleZone | Krull | Qbert |
> | :---        |    :----:   |    :----:   |  :----:   |   :----:   |  :----:   |      ---: |
> | VaGraM  |   648     |  83 |    127    |  4145 | 5686       |  864 |
> | STORM   | 1054        | 177      | 1061       | 7080  |8360       | 2913  |
> | STORM+Var   | 1361        | 248      | 935       | 10140  |8897       | 4243  |
>
>
> The results consistently demonstrate that the modern value-aware method VaGraM performs worse than the maximum likelihood-based approach (STORM) in these high-dimensional visual tasks. This aligns with our hypothesis that without the strong structural prior of RSSM/TSSM (which VaGraM typically lacks), pure value-aware methods struggle to scale to complex visual inputs. We are committed to completing the full benchmark comparisons and will update the final manuscript with the comprehensive results.
>
> ---
>
> **Q3: How does the method conceptually and empirically differ from prior value gradient weighting methods like VaGraM, beyond the obvious architectural differences (RSSM/TSSM vs RNN)?**
>
> **A3:** Beyond the architectural differences (RSSM/TSSM vs. RNN), our method differs fundamentally from VaGraM in its optimization philosophy:
>
> - **VaGraM (Gradient Re-weighting):** VaGraM typically employs sample re-weighting, modifying the importance of data points based on value sensitivity. It uses the value gradient to re-weight the maximum likelihood reconstruction loss. While this approach is more direct, VaGraM relies on first-order approximations to calculate value gradients, which, due to approximation errors, may lead to inaccurate gradients and consequently training instability.
>
> - **Ours (Latent Regularization):** Our method employs latent space regularization. Instead of changing which samples matter, we add a regularization term \$ \mathcal{L}_{var} \$ to constrain the topology of the latent space itself to be value-smooth. Crucially, we retain the dynamics representation objective as a stable anchor. This circumvents the cumbersome and unstable calculation of value gradients for sample weighting, resulting in a more robust training process.
>
> ---

---

> ### Author Response · Authors · 2025-11-25
>
> ---
>
> **Q4: the failures of the proposed method on e.g., PrivateEye for DreamerV3+Var, Frostbite are not analyzed.**
>
> **A4:** We thank the reviewer for identifying this important detail. We have added a new section, **Appendix I: Discussion on Failure Environments**, to analyze this phenomenon.
>
> We hypothesize that the performance drop in PrivateEye and Frostbite stems from their nature as notoriously sparse-reward environments requiring significant initial exploration. In the early training phases of these environments, the value targets are inherently noisy and non-informative (often zero). Forcing the world model to align with such a noisy value function induces overfitting, which drastically reduces the variance in the latent space that is crucial for effective exploration. Unlike Maximum Likelihood Estimation, which naturally encourages capturing all diversity in the observations, premature value alignment in sparse-reward settings can inadvertently hinder the discovery of the first critical reward signal.
>
> ---
>
> **Q5: Have you tried a direct, large-scale comparison? The proposed method regularizes the world model to produce states yielding a similar distribution to real states. Will it lead to delusional states that are easy for the value function to predict but are world model dynamically inconsistent?**
>
> **A5:** This is a critical insight! We have explicitly investigated the potential for "delusional states", that are easy for the value function to predict but are dynamically inconsistent with the environment, in the newly added **Appendix D, Figure 9 (Loss Curves) and Figures 10 & 11 (Visualizations)**.
>
> Figure 9 illustrates the evolution of the dynamics representation reconstruction loss and the value alignment loss on the KungFuMaster environment. The curve analysis confirms that the introduction of value alignment regularization significantly reduces value divergence. However, an excessive bias toward this regularization term, although yielding a lower value alignment loss, concurrently leads to a degradation in the model's fundamental dynamics reconstruction capability.
>
> To visualize this effect, we conducted a further analysis of the trained world models. The results in Visualizations Figure 10 and 11 indicate that when the training excessively favors the value alignment objective, the world model tends to produce delusional states, for example, the enemy's reconstruction exhibits an unstable ghosting artifact in Figure 10, or seemingly materializes instantaneously (spontaneously generated) in Figure 11. Crucially, our designed adaptive weighting mechanism effectively mitigates this instability by dynamically regulating the necessary balance between the dynamics representation reconstruction and the value alignment objectives.

---

### Official Review · Reviewer_TvaB · 2025-10-30

**Soundness:** 2
**Presentation:** 2
**Contribution:** 1
**Rating:** 4
**Confidence:** 4

**Summary:**

The paper proposes Value-aligned World Model Regularization (VAR), which augments a maximum-likelihood world model loss with a KL term between the critic’s intermediate value distribution at real vs. predicted latent states. The method includes a warm-up schedule and an adaptive weighting scheme. Experiments on Atari-100k and DeepMind Control suggest consistent gains over DreamerV3 and STORM, with minimal implementation overhead.

**Strengths:**

- The experiments are extensive and well presented, covering both discrete and continuous control, along with useful ablations on weighting and computational overhead.
- The motivation for “value-aligned world models” is clear, and the concept is explained effectively.
- The paper is generally well written and clearly structured.

**Weaknesses:**

- The paper is heavily centered around previous work (Dreamer-v3) which is totally fine. However to my knowledge, the authors build upon a false assumption that questions a lot of the paper content. By the author's own description of "value aware world models", Dreamer-v3 is indeed a value aware/aligned world model and not just a "classic maximum likelihood" WM. In the original Dreamer-v3 paper (https://arxiv.org/pdf/2301.04104, page 6, first Paragraph) the value alignment is described and can also be found in the original implementation (https://github.com/danijar/dreamerv3, see Agent.py line 219-235 (replay value loss).

- The contribution "we propose Value-aligned World Model, a novel and effective MBRL algorithm" is overstated. Based on my understanding of DreamerV3, the contribution seems to be "yet another way to do value alignment".

- The authors's experiment results for Dreamer-v3 (Table 1) differ significantly from the original paper (https://arxiv.org/pdf/2301.04104, Table 9) in which the models are indeed value-aligned. This seems **not** like a "fair comparison" (see line 371).

- In Appendix, policy entropy is higher than the original, yet claimed to be "consistent" (see Appendix) and "identical" (line 372).

- Figure 1. b) is not discussed or suited w.r.t experiments.

- Equation 4: It is unclear what $\tilde s_t$ is. There is no definition provided.

- The paper repeatedly claims RSSM/TSSM ensures a lower bound while value awareness lifts the upper bound (of what? performance?), but provides no formal definition or guarantee. This reads as metaphor rather than theorem.

- the paper reiterates the Dreamer-v3 theory. It also adds small modifications for, I suspect, "novelty" (e.g. $s_t = (h_t, z_t)$) that seem unnecessary.

**Questions:**

It would be great if the author could address the major weakness I outlined above.

---

> ### Author Response · Authors · 2025-11-26
>
> Dear Reviewer TvaB,
>
> We sincerely appreciate the effort and time dedicated to reviewing our manuscript, as well as your constructive suggestions. In response to your questions, we have prepared the following detailed answers:
>
> ---
>
> **Overall Clarification of the Potential Misunderstanding of the “Value Alignment” Concept**
>
>
> We have thoroughly reviewed your concerns and carefully examined the point you raised regarding *“By the author's own description of 'value aware world models', Dreamer-v3 is indeed a value aware/aligned world model and not just a 'classic maximum likelihood' WM. In the original Dreamer-v3 paper (https://arxiv.org/pdf/2301.04104, page 6, first Paragraph) the value alignment is described and can also be found in the original implementation (https://github.com/danijar/dreamerv3, see Agent.py line 219-235 (replay value loss).”*
>
> Here, we wish to formally and respectfully clarify that the "value alignment" mechanism mentioned in the DreamerV3 paper (Page 6, first Paragraph) is **fundamentally different** from our proposed Value Alignment Regularization, differing entirely in motivation, design objective,  optimization function, and practical implementation. We believe there is a **significant technical misunderstanding** regarding these two concepts.
>
> - In standard Temporal Difference (TD) Q-learning, the simultaneous real-time updates of the main network \$Q\$ inevitably alter the target \$ max_a Q(s', a) \$, creating an unstable self-referential loop that often leads to training oscillations or divergence. The "value alignment" described in the page 6, first paragraph of the Dreamer-v3 paper is more accurately characterized as a common optimization technique designed to stabilize the training of the value network. Its primary goal is to fix the Bellman TD target; for instance, DQN achieves this via periodically cloning the value network every N steps, while Dreamer-v3 employs an exponentially moving average (EMA) on the value network's parameters to create a "soft" Target Network for stable target generation. Crucially, this stability technique is applied exclusively during the agent’s value learning phase, remaining completely decoupled from the dynamic learning of the world model, and does not actively guide the world model to focus on any value-sensitive information. Therefore, it is fundamentally an optimization trick at the value learning level. Of course, this optimization strategy does not conflict with our proposed method, and we also implement this strategy in our code to prevent training instability during value network learning.
>
> - In sharp contrast, the core of the value alignment we propose focuses on the training of the world model. Our goal is to incorporate a value alignment regularization as an inductive bias by leveraging the task importance naturally encoded in the value function. This regularization constrains the world model's learned latent state distribution to align with the task-critical features captured by the value network. Unlike the aforementioned methods, we do not perform operations in the parameter space, nor are we aiming to stabilize value training. Instead, we directly use the trained value network to guide the world model to focus on value-relevant information in the latent space through KL divergence.
>
> Based on the analysis above, DreamerV3 world model training relies solely on maximum likelihood estimation. It lacks value guidance in both architecture design and loss optimization, therefore is not what we describe as the value-aware world model. Meanwhile, the contribution of our paper should not be reduced to "seems to be 'yet another way to do value alignment'." In contract, we introduce a Value-alignment regularization term into the maximum likelihood world model optimization, allowing the world model to not only focus on modeling environmental dynamics but also prioritize the reconstruction of states with high value sensitivity.
>
> ---
>
> **Q3: The authors's experiment results for DreamerV3 (Table 1) differ significantly from the original paper (https://arxiv.org/pdf/2301.04104, Table 9) in which the models are indeed value-aligned. This seems not like a "fair comparison" (see line 371).**
>
> **A3:** We have carefully re-examined the literature and identified the source of this discrepancy, which stems from specific version configurations used in the original DreamerV3 paper. The detailed analysis can be referred to Reviewer MsDe's Q1&A1.
>
> ---

---

> ### Author Response · Authors · 2025-11-26
>
> ---
>
> **Q4: In Appendix, policy entropy is higher than the original, yet claimed to be "consistent" (see Appendix) and "identical" (line 372).**
>
> **A4:** In practice, we consistently use a policy entropy coefficient of $3 \times 10^{-4}$ on the Atari 100K benchmark, while employing a larger coefficient of $1 \times 10^{-3}$ on the DMC suite to enhance exploration. We will add this description in our final manuscript.
>
> ---
>
> **Q6: Equation 4: It is unclear what $ \\tilde{s}_{t} $ is. There is no definition provided.**
>
> **A6:** Thank you for your careful observation! $\tilde{s}_t$ refers to $[h_t, \tilde{z}_t]$, and we will update this consistently in the final manuscript.
>
> ---
>
> **Q7: The paper repeatedly claims RSSM/TSSM ensures a lower bound while value awareness lifts the upper bound (of what? performance?), but provides no formal definition or guarantee. This reads as metaphor rather than theorem.**
>
> **A7:** We thank the reviewer for this constructive suggestion! In newly added **Appendix J**, we have clarified that the concepts of "lower" and "upper" bounds serve as conceptual metaphors for the model's performance capabilities:
>
> - **Lower Bound (Fidelity & Validity):** This refers to the model's fidelity to the physical simulation. By employing the RSSM/TSSM architecture to learn dynamic representations guarantees that the learned representations are at least physically consistent. This prevents the model from collapsing into hallucinations that completely deviate from the ground-truth environment dynamics.
>
> - **Upper Bound (Utility & Optimality):** This refers to the model's utility for a specific task. Under finite capacity constraints, a pure MLE approach often wastes capacity on high-entropy but task-irrelevant backgrounds. Introducing Value-Alignment regularization acts as an inductive bias that rectifies this focus, thereby raising the performance ceiling (upper bound) by ensuring the representation captures task-critical features.
>
>
> Once again, we thank you for your time, thoughtful insights and constructive suggestions. Should you have any further questions or additional feedback, please feel free to let us know.

---

> > ### Comment · Reviewer_TvaB · 2025-11-27
> >
> > I thank the authors for their detailed feedback! However I think, my major concern is still not properly adressed.
> > Your answer to my concern reformulates the desription from the next paragraph (Because the critic regresses targets that depend […]) which is not the issue.
> >
> > To my understanding, the gradient of the replay value loss from the original Hafner implementation goes into the representation model!
> > This is why in the Dreamerv3 paper it is stated that "[...] To improve value prediction in environments where rewards are challenging to predict, we apply the critic loss both to imagined trajectories [...] and to trajectories sampled from the replay buffer". This is the purpose of the replay value loss.
> > Consequently, the representation model (which gets gradients from replay value loss) is used by the world model dynamics so this means that Dreamerv3 is indeed a value-aligned world model.
> > Therefore, it acts as an inductive bias like your method. I would be happy if the authors would adress this explicitly.
> >
> >
> > In addition, I could not find a reference to the PyTorch implementation that you used. So, I assumed this one: https://github.com/NM512/dreamerv3-torch, where I could not find the replay value loss.
> > It would be very helpful if the authors could give a reference to the replay value loss in their PyTorch implementation.

---

> > > ### Author Response · Authors · 2025-11-28
> > >
> > > Thank you for your timely response and constructive suggestions! I now fully understand your concerns. I have carefully reviewed the statement in the original DreamerV3 paper:
> > >
> > > *“To improve value prediction in environments where rewards are challenging to predict, we apply the critic loss both to imagined trajectories with loss scale $β_{val}$ = 1 and to trajectories sampled from the replay buffer with loss scale $β_{repval}$ = 0.3. The critic replay loss uses the imagination returns $R^λ_t$ at the start states of the imagination rollouts as on-policy value annotations for the replay trajectory to then compute λ-returns over the replay rewards.”*
> > >
> > > Additionally, I examined the official code implementation by the authors (https://github.com/danijar/dreamerv3, see Agent.py line 219-235 (replay value loss)).
> > >
> > > ---
> > >
> > > **A1: DreamerV3 indeed not a value-aligned world model**
> > >
> > > In the official code at Agent.py line 220, `feat = sg(repfeat, skip=self.config.repval_grad)`, where `repfeat` represents the trajectories from the replay buffer, and `sg` refers to the stop gradient operation. From Hafner’s official code, it is evident that the gradient is effectively truncated by the stop gradient operation, meaning that **the gradient of the replay value loss from the original Hafner implementation does not propagate into the representation model**.
> > >
> > > Therefore, based on the authors' original statement, the purpose of adding the replay value loss is to **improve value prediction**, as the authors believe that using only imagined trajectories could result in value misalignment in environments with sparse or difficult-to-predict rewards. The additional trajectories from real simulation samples are used to optimize the value network, which can be more accurately understood as a data augmentation technique rather than a value alignment approach.
> > >
> > > **Based on the above analysis, I believe it can be clearly clarified that DreamerV3 is not the value-aligned world model as we described.** The authors of DreamerV3 use the stop gradient operation to truncate the gradient flow of the value loss to the representation model, meaning that the world representation model is not updated during the training of the agent's actor and critic. In our view, truncating the gradient flow to the representation model is intuitive because without this truncation, the training process would involve multiple coupled network parameters being updated simultaneously, leading to instability in the overall training process.
> > >
> > > ---
> > >
> > > **A2: Practical implementation and subsequent ablation experiments**
> > >
> > > We did indeed use the DreamerV3 implementation in PyTorch from [https://github.com/NM512/dreamerv3-torch](https://github.com/NM512/dreamerv3-torch). As we described in our response to Reviewer MsDe’s Q1&A1, DreamerV3 has two versions: arXiv v1 (https://arxiv.org/abs/2301.04104v1) and arXiv v2 (https://arxiv.org/pdf/2301.04104v2). The PyTorch implementation of DreamerV3 primarily follows the v1 version. After carefully reviewing the v1 version, we found that the authors did not mention the replay value loss in v1, so it is not implemented in the PyTorch version either.
> > >
> > > As we analyzed above, the replay value loss is merely a data augmentation strategy for value training. It introduces real interaction trajectories, in addition to imagined trajectories, to improve value prediction. While I am not entirely sure whether this approach violates the principle that the agent should only train through virtual interactions with the world model, I firmly believe that even with the introduction of replay value loss, it will not affect our conclusions. In fact, improving the value network’s predictions could actually benefit the value alignment in world model training.
> > >
> > > We also noticed that Reviewer CzbV similarly mentioned, *"Comparison and ablation studies with different value alignment methods (e.g., making value functions differentiable during actor-critic training with gradients flowing through the state) would clarify this."* Removing the stop gradient operation in the official implementation of DreamerV3 would indeed allow us to achieve this, and this could be considered a way to introduce value alignment. We are currently conducting related ablation experiments, and once the experimental results are ready, we will promptly provide feedback!
> > >
> > > We greatly appreciate your time and effort in reviewing our work and look forward to your further response. We hope that, through our joint efforts, we can further improve our manuscript to a better version.

---

> > > > ### Comment · Reviewer_TvaB · 2025-11-28
> > > >
> > > > I thank the authors again for the detailed response!
> > > >
> > > > Line 220 in agent.py is indeed the crucial line to consider! Here, we need to look at the argument `skip=self.config.repval_grad`.
> > > > If the argument is True, then the stop-gradient is not applied and just skipped resulting in gradient flow to the representation model (`sg` operation is defined in agent.py line 17).
> > > > When you look in the config file (https://github.com/danijar/dreamerv3/blob/main/dreamerv3/configs.yaml), lines 115 and 116 show that
> > > > `repval_loss: True`, `repval_grad: True`. Therefore DreamerV3 is a value-aligned world model.
> > > >
> > > > It would be very helpful if the authors could adress this concern explicitly again!

---

> ### Author Response · Authors · 2025-12-03
>
> We sincerely appreciate the reviewer’s timely feedback and constructive suggestions. We agree that in the most recent version of DreamerV3 (v2), the authors introduced a form of value-awareness through the use of a replay value loss, enabling value conditioning during training. In contrast, prior versions, namely DreamerV1, DreamerV2, and the initial release of DreamerV3 (v1), did not incorporate such mechanisms. We will clarify and update this distinction in our revised manuscript to ensure accurate attribution.
>
> However, our proposed value-alignment strategy differs significantly from the replay value loss mechanism. To better illustrate these differences, we provide comparative experiments (summarized in the table below) evaluating both methods, applying the replay value loss to DreamerV3 and STORM, respectively.
>
> | Atari 100K    | Alien | Amidar | CrazyClimber | Gopher | KungFuMaster | Qbert |
> | :---        |    :----:   |    :----:   |  :----:   |   :----:   |  :----:   |      ---: |
> | DreamerV3  |   876     |  144 | 89900       |  1684 | 16375      |  1224 |
> |DreamerV3+replay value loss|748|129|85245|1876|20425|2123|
> |DreamerV3+Var(Ours)|1233|185|81720|2807|21075|2268|
> | STORM   | 1054        | 177      | 47473      | 8551  | 15760      | 2913  |
> |STORM+replay value loss|861 |199|46068|3173|20558|4146|
> | STORM+Var(Ours)   | 1361        | 248      | 57335       | 13510  | 26615      | 4243  |
>
> The experimental results indicate that while the replay value loss method can improve model performance in certain environments(KungFuMaster, Qbert), it also leads to degradation in others(Alien, Amidar, Gopher), resulting in inconsistent outcomes. In contrast, our method, which directly aligns the prior and posterior latent distributions of the world model using value guidance, achieves more consistent and significant improvements across environments. We believe the core difference lies in the nature of their optimization objectives: whereas the gradient-based method primarily focuses on reducing the value prediction error, our approach explicitly minimizes the distributional discrepancy between the prior and posterior latent states, making it more aligned with the fundamental optimization goal of world model learning.

---

### Official Review · Reviewer_CzbV · 2025-10-30

**Soundness:** 2
**Presentation:** 2
**Contribution:** 2
**Rating:** 4
**Confidence:** 4

**Summary:**

This paper addresses a common limitation in model-based reinforcement learning, where world models trained via maximum likelihood on reconstruction objectives often expend model capacity on pixel-level details that are irrelevant to the downstream task. This can lead to state representations full of redundant information.

The authors propose a value-alignment regularization term that is added to the standard MLE objective. This new loss term is designed to make the world model task-aware by aligning its learned representations with the agent’s value function. Specifically, the regularization loss minimizes the KL divergence between the value predicted from the model's imagined latent state and the value predicted from the latent state inferred from observation. By integrating this value-based loss directly into the world model's training, the model is explicitly encouraged to prioritize learning and representing state features that are useful for predicting future returns, rather than focusing solely on reconstructing all sensory details. This additional loss term can be added to any supervised model learning problem within an RL setting.

**Strengths:**

- interesting idea with easy integration into different maximum likelihood MBRL approaches, possibly contributing to a better understanding of what aspects of environment interactions / agent experience carry meaningful information for good world models
- helps with task-specific world models by forcing the world model to align with the actual problem (potentially filtering out unnecessary information to solve the task)
- experiments show improved performance on most tasks

**Weaknesses:**

- the approach seems very similar to the representational learning approach in MuDreamer. to judge the originality of the present approach it would be important to know how the authors position themselves wrt this prior work.
- not well motivated: why is the choice of regularization, namely matching imagined values vs. grounded values, reasonable? The analogy to “perceptual losses” in computer vision is mentioned but not explained.
- warm-up seems a bit arbitrary, adaptive weight mechanisms to prevent instabilities during early training introduces additional hyper-parameters
- missing analysis of the actual learnt representations, mostly performance tests. Figure 3 tries to achieve this but leaves a lot of room for interpretation. Focus was on performance on the benchmark task, not on understanding the representational implications of the approach.
- Figure 2 compares performance for a) different model sizes and b) different observation sizes but only for 100k steps. The results may be interpreted as the proposed approach being more sample efficient but none of the models are trained to full convergence, making statements about performance (model captures dynamics better / worse) inconclusive.
- Figure 1 does not help much for understanding

**Questions:**

- performance improvements seem much more pronounced in the case of ATARI, but a lot weaker for DMC. Why is that? This raises questions about the general applicability of the approach
- How does $\beta_{var}$  behave over time? Do the two paradigms align or are they counteracting each other, resulting in alternating optimization between maximum likelihood optimization and value alignment?
- Representation power? do the learned representations actually align with the task? something clearer than Fig 3
 How does this approach compare to MuDreamer, which seems to follow a similar idea?

---

> ### Author Response · Authors · 2025-11-25
>
> Dear Reviewer CzbV,
>
> We sincerely appreciate your detailed feedback and constructive suggestions, which have helped us significantly clarify the positioning and mechanics of our work. We have addressed your concerns point-by-point below:
>
> ---
>
> **Q1: the approach seems very similar to the representational learning approach in MuDreamer. How does this approach compare to MuDreamer, which seems to follow a similar idea?**
>
> **A1:** We thank the reviewer for this insightful observation! While MuDreamer and our work share a similar motivation, driving the world model to focus on task-relevant features, the two methodologies are fundamentally distinct in their mechanism, optimization philosophy, and practical implementation.
>
> **(1) Distinct Core Insights (Reconstruction-Free vs. Regularized Reconstruction)**
>
> - MuDreamer adopts a "subtraction" approach. It posits that pixel reconstruction is unnecessary, as it forces the world model to learn irrelevant details (e.g., background). Consequently, it removes the decoder entirely, relying solely on value/reward prediction to shape the latent space.
>
> - Conversely, we adopt an "addition" (hybrid) approach. We argue that pixel reconstruction is necessary as it provides fundamental training stability (guaranteeing the world model's fidelity to environmental dynamics and preventing complete divergence). We retain the reconstruction objective while adding Value Alignment Regularization to guide the attention toward task-critical features.
>
>  **(2) Distinct Optimization Objectives (Indirect vs. Direct Latent Shaping)**
>
> - MuDreamer eliminates the pixel reconstruction loss, instead utilizing the auxiliary prediction heads (value, reward, action) to indirectly shape the latent space via prediction error minimization.
>
> - We applies value alignment directly to the latent space distribution via KL-divergence. Crucially, we introduce an adaptive weighting mechanism to explicitly balance the reconstruction fidelity provided by MLE and the task-relevance provided by value alignment.
>
> **(3) Distinct Technical Implementation & Stability**
>
> - Because MuDreamer abandons reconstruction, it faces the risk of representation collapse. To mitigate this, it necessitates specific architectural constraints, such as the critical addition of Batch Normalization to the representation network.
> - By retaining reconstruction, our method naturally avoids collapse without requiring such architectural changes. Our approach is designed as a plug-and-play module that can be seamlessly integrated into existing MLE-based frameworks (like DreamerV3 and STORM) with minimal code changes.
>
> We thank the reviewer for pointing out the relevant work, MuDreamer. We will definitely include it and discuss these differences in the "Related Work" section in our final revision.
>
> ---
>
> **Q2: not well motivated: why is the choice of regularization, namely matching imagined values vs. grounded values, reasonable? The analogy to “perceptual losses” in computer vision is mentioned but not explained.**
>
> **A2:** The reviewer mentions "matching imagined vs. grounded values." We believe a more precise characterization of our objective is "matching the prior value distribution vs. the posterior value distribution."
>
> Standard Maximum Likelihood Estimation treats every pixel equally, often wasting capacity on high-entropy but task-irrelevant backgrounds (distractors). The value function, however, intrinsically encodes task importance, only state features that influence expected returns cause value changes. Therefore, forcing the prior (predicted state) to match the value distribution of the posterior (observed state) drives the latent space to prioritize features that matter for the task.
>
> **Analogy to Perceptual Loss:** In computer vision, Perceptual Loss uses high-level features from a pre-trained network to guide image generation, avoiding the pitfalls of pixel-wise MSE (which leads to blurriness). Similarly, we leverage the Value Network as a "pre-trained evaluator." Its distribution serves as a high-level semantic supervision signal, compelling the world model to produce latent states that are accurate in "value space" (task-relevant features) rather than just "pixel space" (visual features).
>
> ---

---

> ### Author Response · Authors · 2025-11-25
>
> ---
>
> **Q3: missing analysis of the actual learnt representations. Focus was on performance on the benchmark task, not on understanding the representational implications of the approach. do the learned representations actually align with the task?**
>
> **A3:** We thank the reviewer for this suggestion. To address this, we have added a detailed task-level analysis in newly added **Appendix F, Figure13 (Detailed Analysis of World Model Visualization).**
>
> Taking the KungFuMaster environment as a case study, the core objective is to control the central agent to defeat enemies approaching from both the left and right sides to clear the stage. Consequently, the critical challenge in this environment lies in accurately reconstructing the agent and the enemies, whereas the static background information can be selectively ignored.
>
> From the visualization results in Figure 13 of  Appendix F, two key observations can be made:
>
> - **Decoded Reconstructions:** Vanilla STORM often fails to reconstruct key entities (e.g., the enemies are blurry or missing, as highlighted by red circles). Introducing Value Alignment forces the model to retain these critical features, resulting in sharper reconstructions of the key entities.
>
> - **Saliency Maps:** The value-weighted saliency maps demonstrate that, unlike the baseline, our value-aligned model focuses its attention intensity specifically on the controlled character and incoming enemies, effectively suppressing the static background. This empirically confirms that the learned representations are indeed aligned with the task.
>
> ---
>
> **Q4: Figure 2 compares performance for a) different model sizes and b) different observation sizes but only for 100k steps. The results may be interpreted as the proposed approach being more sample efficient but none of the models are trained to full convergence, making statements about performance (model captures dynamics better / worse) inconclusive.**
>
> **A4:** To validate our method's potential under model capacity constraints at convergence, we conducted new experiments on the DeepMind Control Suite (Quadruped Run), extending training to 1 Million steps to ensure convergence (detailed discussion see **Appendix H, Figure 14**).
>
> Figure 14 illustrates the training curves for both the small and medium model sizes. Under the medium model size, the introduction of value alignment regularization yields performance comparable to the original DreamerV3 algorithm. However, when the model size is reduced, the baseline DreamerV3 suffers a significant performance drop as it struggles to model the complex dynamics with limited parameters. However, introducing Value Alignment effectively mitigates this drop, significantly raising the performance ceiling.
>
> ---
>
> **Q5: performance improvements seem much more pronounced in the case of ATARI, but a lot weaker for DMC. Why is that? This raises questions about the general applicability of the approach**
>
> **A5:** Good question! The observed difference in performance gain, substantial improvement on Atari environments versus smaller improvement on the DMC suite, actually validates the mechanism of our proposed method:
>
> - **Atari Environments (High Gain):** Atari games typically involve high-dimensional visual inputs with heavy distractors (complex backgrounds) and small critical objects. This causes Maximum Likelihood-based world models to waste representational capacity on reconstructing task-irrelevant backgrounds. In this scenario, introducing Value-Alignment Regularization provides a strong inductive bias to filter noise, leading to substantial gains.
>
> - **DMC Environments (Lower Gain):** As the reviewer MsDe noted, DMC tasks often use proprioceptive inputs or clean visuals where "all state dimensions are task-relevant." In these high-signal-to-noise environments, the standard MLE model is already sufficient to capture the dynamics. Consequently, the marginal utility of Value Alignment naturally diminishes.
>
> ---

---

> ### Author Response · Authors · 2025-11-25
>
> ---
>
> **Q6: How does \$\beta_{var}\$ behave over time? Do the two paradigms align or are they counteracting each other, resulting in alternating optimization between maximum likelihood optimization and value alignment?**
>
> **A6:** We investigated the trajectory of the adaptive weight \$\beta_{var}\$ in newly added **Appendix E, Figure 12**. Figure 12 illustrates the change curves of the dynamics representation reconstruction loss, the value alignment loss, and the adaptive weight across various Atari games.
>
> The results indicate that in the initial learning stage, the dynamics representation loss is high, and the adaptive weight is decreasing, meaning the primary focus is on Maximum Likelihood Reconstruction. As training stabilizes, the dynamics representation loss decreases, and the adaptive weight increases, signaling a shift in attention toward task-critical features.
>
> Crucially, the curves show that as \$\beta_{var}\$ rises, the value alignment loss drops significantly while the dynamics representation loss remains stable. This demonstrates that the two objectives are synergistic, not antagonistic. The model learns to capture value-relevant features without sacrificing its ability to model the environment's dynamic.
>
> ---
>
> **Q7: warm-up seems a bit arbitrary, adaptive weight mechanisms to prevent instabilities during early training introduces additional hyper-parameters.**
>
> **A7:** On the one hand, warm-up is helpful because the value network is randomly initialized at the start. Aligning to it immediately would inject noise into the latent space. On the other hand, contrary to introducing complexity, the adaptive weight actually obviates the need for manual hyperparameter tuning. By defining adaptive weight as a function of the dynamics loss, we create an automatic curriculum: the weight self-regulates based on the quality of the world model. When the model is confused (high reconstruction error), it turns off regularization; when the model is confident, it turns on alignment.

---

> > ### Comment · Reviewer_CzbV · 2025-11-27
> >
> > We thank the authors for their clarifications, additional experiments and responses. While most of the minor issues were resolved, the core question of why this approach was chosen in comparison to other possibilities (e.g. MuDreamer, independent of the reconstruction issue) remains unclear. Comparison and ablation studies with different value alignment methods (e.g. making value functions differentiable during actor critic training with gradients flowing through the state) would clarify this. Similarly, why is this approach limited to value functions and could not equally be applied to reward prediction, action prediction, continuity prediction, etc.
> >
> > The concerns regarding anecdotal representation analysis remain. It seems Figure 13 is simply a magnified version of Figure 3 with red circles. It is really hard to tell a meaningful difference between the approaches. It looks like both approaches suffer from ghosting when looking at different frames. The saliency map looks mostly random and seems to suggest that the most far away enemy has the highest attention (last frame), etc.

---

> > > ### Author Response · Authors · 2025-12-03
> > >
> > > We sincerely thank the reviewer for their time and effort! In response to your concerns, we provide detailed point-by-point replies below:
> > >
> > > We agree that comparing against alternative value alignment methods is critically important. To provide a more comprehensive evaluation, we have conducted the following additional experiments:
> > >
> > > | Atari 100K    | Alien | Amidar | CrazyClimber | Gopher | KungFuMaster | Qbert |
> > > | :---        |    :----:   |    :----:   |  :----:   |   :----:   |  :----:   |      ---: |
> > > | DreamerV3  |   876     |  144 | 89900       |  1684 | 16375      |  1224 |
> > > |DreamerV3+Gradient flow|748|129|85245|1876|20425|2123|
> > > |DreamerV3+Var(Ours)|1233|185|81720|2807|21075|2268|
> > > | STORM   | 1054        | 177      | 47473      | 8551  | 15760      | 2913  |
> > > |STORM+Gradient flow|861 |199|46068|3173|20558|4146|
> > > | STORM+Var(Ours)   | 1361        | 248      | 57335       | 13510  | 26615      | 4243  |
> > >
> > > - As illustrated in the above table, we compare our proposed method with an alternative approach that makes value functions differentiable during actor critic training with gradients flowing through the state. We evaluate both DreamerV3 and STORM, representing RSSM- and TSSM-based frameworks respectively, using five random seeds for robustness. The experimental results indicate that while this gradient-based alignment method can improve model performance in certain environments, it also leads to degradation in others, resulting in inconsistent outcomes. In contrast, our method, which directly aligns the prior and posterior latent distributions of the world model using value guidance, achieves more consistent and significant improvements across environments. We believe the core difference lies in the nature of their optimization objectives: whereas the gradient-based method primarily focuses on reducing the value prediction error, our approach explicitly minimizes the distributional discrepancy between the prior and posterior latent states, making it more aligned with the fundamental optimization goal of world model learning.
> > >
> > > | Atari 100K    | Alien | KungFuMaster |
> > > | :---        |    :----:   |    :----:   |
> > > | DreamerV3  |   876     | 16375      |
> > > |DreamerV3+reward\&continuity alignment|1026|19388|
> > > |DreamerV3+Var(Ours)|1233|21075|
> > >
> > > - To assess the generality of our alignment strategy, we further apply the same alignment mechanism to alternative predictive heads such as reward and continuity signals. The results suggest that while these variants do yield marginal improvements over the baseline, they fall short of the performance gains achieved through value alignment. This reinforces the conclusion that value functions offer more effective task-relevant supervision for guiding representation learning within the world model.
> > >
> > > | KungFuMaster   | MSE | value error  |
> > > | :---        |    :----:   |    :----:   |
> > > | STORM  |  $ 3.2 \times 10^{-4} $    | 111.8      |
> > > |STORM+Var(Ours)|$ 2.7 \times 10^{-4} $|38.9|
> > >
> > > -  Additionally, we quantitatively evaluate the learned world model by comparing imagined trajectories with real environment trajectories in the KungFuMaster environment. We collect a total of 3,120 sequences and compute both the Mean Squared Error (MSE) and the value prediction error between imagined and ground-truth rollouts. The results demonstrate that the value-aligned model significantly outperforms the baseline in terms of both absolute error and task-relevant value estimation, providing strong empirical evidence for the effectiveness of our alignment mechanism in enhancing the fidelity and utility of imagined rollouts.

---

### Note · Authors · 2026-01-27

I have read and agree with the venue's withdrawal policy on behalf of myself and my co-authors.

---

### Meta-Review · Area_Chair_ujMd · 2026-01-06

**Summary:**

This paper proposes to introduce a value-alignment regularization term in the learned dynamics to improve the learned representation.

Strengths:
- Reviewers appreciate its simple and easy to implement approach
- Performance improvement on most tasks

Main concerns:
1. (TvaB, MsDe) Discrepancy in the reported score of baselines (DreamerV3 and STORM) from the original paper
2. (CzbV) Motivation on the specific choice of value-alignment, comparison to alternatives
3. (CzbV) In Figure 2, none is trained to full convergence, leading to inconclusive statements
4. (CzbV, Hona) Lack of analysis of the learned representation, risk of learning delusional states
5. (TvaB) Dreamer-v3 is indeed a value aware/aligned world model rather than a pure maximum likelihood model
6. (TvaB, Hona) Unjustified statements about the lower bound from RSSM/TSSM and upper bound from value awareness
7. (Hona) Lack of comparison with modern value-aware methods such as VaGraM or CVAML and lack of analysis on failures

**Reviewer Concerns:**

Concern 1: the authors points out that the discrepancy is due to a different version of DreamerV3 paper (v1 vs v2) and training randomness in STORM. Recent literatures also use v1 (worse and smaller model) for comparison. However, this leaves the question open whether the proposed model can improve the more power algorithm (DreamerV3 v2)

Concern 2: the authors provided more explanation on the design choice but reviewer CzbV remained skeptical about its advantage compared to alternatives. The authors further provided experiments to evaluate a few options (gradient flowing through the state, alignment on predicted reward and continuity signals).

Concern 3. The rebuttal includes new experiments with 1M steps. It shows the proposed method achieves better performance than the baseline with a small model size, but becomes comparable when increasing to a medium model size. The results remain inconclusive.

Concern 4. The authors provided additional analysis on the learned representation. It should address the concern from Hona about learning delusional states but CzbV remained skeptical on the interpretation of the learned representation

Concern 5. The authors admit Dreamer-v3 v2 is a value aware world model after discussion. The v1 code is used in this submission. This confirms the concern from reviewer TvaB.

Concern 6. The authors explain the mentions of "lower bound" and "upper bound" serve as conceptual metaphors for the model's performance capabilities. It corroborate the initial concern and requires a more rigorous explanation.

Concern 7. The rebuttal provides additional comparison to the baselines and discussion on the failure cases. This concern should have been resolved.

**Reviewer Scores:**

CzbV, initial 4, predicted 4 or 5. Although the majority of minor concerns are resolved, the main ones about the value alignment design choice and the interpretation of learned representation remain.

TvaB, initial 4, predicted 4 or 3. The author-reviewer discussion confirms main concerns from this reviewer.

Hona, initial 4, predicted 5 or 6. The concerns about comparison with recent baselines and analysis of failure cases and representation learning are resolved, but concern 6 remains.

MsDe, initial 4, raised to 8 after discussion. Likely to stay at 8 after rebuttal.

---

### Decision · Program_Chairs · 2026-01-26

Reject